# InfoNCE is a variational autoencoder

## Abstract

There are two main approaches to self-supervised learning (SSL), generative SSL, where we learn a full probabilistic model of all the inputs, or contrastive SSL where we train on a supervised learning task that has been carefully designed to encourage good representations. We reconcile generative and contrastive SSL by showing that contrastive SSL methods (including InfoNCE) which are motivated in terms of maximizing the mutual information (MI) implicitly learn a full probabilistic model of the inputs, parameterised as a variational autoencoder (VAE). In particular, when we learn the optimal prior, the VAE objective (the ELBO) becomes equal to the MI. In turn, we show that for a deterministic encoder the ELBO is equal to the log Bayesian model evidence. This establishes a profound connection between Bayesian inference and information theory. However, practical InfoNCE methods do not use the MI as an objective: the MI is invariant to arbitrary invertible transformations, so using an MI objective can lead to highly entangled representations (Tschannen et al., 2019). Instead, the actual InfoNCE objective is a simplified lower bound on the MI which is loose even in the infinite sample limit. This raises an important question: does it really make sense to motivate an objective that works (i.e. the actual InfoNCE objective) as a loose bound on an objective that does not work (i.e. the true MI which gives arbitrarily entangled representations)? We give an alternative motivation for the actual InfoNCE objective. In particular, we show that in the infinite sample limit, and for a particular choice of prior, the actual InfoNCE objective is equal to the log Bayesian model evidence. Thus, we argue that it make sense to motivate the InfoNCE from our VAE perspective (as the actual InfoNCE objective is equal to the log Bayesian model evidence), as opposed to the MI perspective (as the actual InfoNCE objective forms only a loose bound on the MI).

## 1 Introduction

A common challenge occuring across machine learning is to extract useful, structured representations from unlabelled data (such as images). This problem is known as self-supervised learning, and there are two broad approaches: generative and contrastive (Liu et al., 2021).

Generative self-supervised learning (also known as unsupervised learning) can be traced back at least to the Boltzmann machine (Ackley et al., 1985) and the Helmholtz machine (Dayan et al., 1995; Hinton et al., 1995). This classical work emphasises two key characteristics of most generative models; first, they should in some sense model the probability density of the data and second they should use latent variables that are ideally interpretable. Modern generative models are exemplified by variational autoencoders (VAEs) (Kingma & Welling, 2013; Rezende et al., 2014). VAEs (like the Helmholtz machine) learn a probabilitistic encoder which maps from the data to a latent representation, and learn a decoder which maps from the latent representation back to the data domain. This highlights perhaps the key issue with VAEs: the need to reconstruct the data, which may be highly complex (e.g. images) (Dorta et al., 2018) which may force the latent space to encode details of the image that are irrelevant for forming a good high-level representation (Chen et al., 2016b).

Contrastive self-supervised learning is an alternative class of methods that learn good representations without needing to reconstruct the data. One common approach is to define a "pretext" classification task (Dosovitskiy et al., 2015; Noroozi & Favaro, 2016; Doersch et al., 2015; Gidaris et al., 2018). For instance,

we might take a number of images, rotate them, and then ask the model to determine the rotation applied (Gidaris et al., 2018). The rotation can be identified by looking at the objects in the image (e.g. grass is typically on the bottom of the image, while birds are nearer the top), and thus a representation useful for determining the orientation may also extract useful information for other high-level tasks. We are interested in an alternative class of objectives known as InfoNCE (NCE standing for noise contrastive estimation) (Oord et al., 2018). These methods take two inputs (e.g. two different patches from the same underlying image), encode them to form two latent representations, and use a classification task to maximize a bound on the mutual information between them. As the shared information should concern high-level properties such as objects, but not low-level details of each patch, this should again extract a useful representation.

InfoNCE was thought to learn good representations by (approximately) maximizing mutual information (Oord et al., 2018). However, recent work has argued that maximizing the true mutual information could lead to arbitrarily entangled representations, as the mutual information is invariant under arbitrary invertible transformations (Tschannen et al., 2019). Instead, they argue that InfoNCE learns good representations because it uses a highly simplified lower bound on the information estimator (Oord et al., 2018) which forms only a loose bound on the true MI. This is highly problematic: Tschannen et al. (2019) argue that better MI estimators give worse representations (Tschannen et al., 2019), so InfoNCE's success with a highly approximate estimator cannot be due to maximizing mutual information, but instead appears to be due to an ad-hoc choice of simplified mutual information estimator. So what is InfoNCE doing? And how can the success of its simplified mutual information estimator be understood?

Here, we develop a new family of contrastive self-supervised variational autoencoders (CSSVAEs). The CSSVAE objective is the ELBO. Remarkably, we show that the CSSVAE ELBO is equivalent to the log-Bayesian model evidence in the usual case where the encoders are deterministic (Sec. 3.3). Further, we show that the log-Bayesian model evidence is equal to the MI under an optimized prior (Sec. 3.5), and equal to the infinite-sample limit of the InfoNCE objective under a different choice of prior (Sec. 3.6). As such, our work has important implications for the interpretation of the InfoNCE objective. In particular, the InfoNCE objective forms only a loose bound on the true MI, even in the infinite sample setting (Sec. 2.2). In contrast, in the infinite sample setting, the InfoNCE objective is equal to the log model evidence. This would argue that the InfoNCE objective is better motivated in terms of the log-Bayesian model evidence (as they are equal in the infinite sample setting), as opposed to the mutual information (as the InfoNCE objective only forms a bound in the same setting). As such, we unify contrastive and generative SSL. Finally, we highlight that the generative model viewpoint is useful for designing new InfoNCE-like learning methods in terms of priors in the latent space. We give an example in a toy system in which the usual choice of InfoNCE objective fails completely, but a modified system that exploits our prior knowledge about dynamics in the latent space succeeds (Sec. 4).

## 2 Background

### 2.1 Variational inference (VI) and variational autoencoders (VAE)

Variational autoencoders (Kingma & Welling, 2013; Rezende et al., 2014) are a special case of VI. In VI, we have observed data, $x$, and latents, $z$, and we specify a prior, $P_\theta(z)$, a likelihood, $P_\theta(x|z)$, and an approximate posterior, $Q_\phi(z|x)$. We then jointly optimize parameters of the generative model, $\theta$, and approximate posterior, $\phi$, using the ELBO,

$$\log P_\theta(x) \geq \mathcal{L}(x; \theta, \phi) = E_{Q_\phi(z|x)}\left[\log \frac{P_\theta(x|z)P_\theta(z)}{Q_\phi(z|x)}\right], \tag{1}$$

which bounds the model evidence or marginal likelihood, $P_\theta(x)$ (as can be shown using Jensen's inequality). The only difference between a VAE and a generic VI strategy is the form of the approximate posterior, $Q_\phi(z|x)$. In a VAE, the approximate posterior is parameterised by a neural network that maps the datapoint, $x$, into a distribution over $z$. Such an approximate posterior, is often known as an encoder as it maps from data to latents, while the likelihood, $P_\theta(x|z)$, is often known as the decoder, as it maps from latents back to the data domain. In contrast, general VI allows for alternative parameterisations of the approximate posterior. For instance, we could explicitly learn a Gaussian mean for each datapoint separately.

We can rewrite the ELBO as an expected log-likelihood plus a KL-divergence,

$$\mathcal{L}(x; \theta, \phi) = \mathrm{E}_{\mathrm{Q}_\phi(z|x)} \left[ \mathrm{P}_\theta \left( x|z \right) \right] - \mathrm{D}_{\mathrm{KL}} \left( \mathrm{Q}_\phi \left( z|x \right) \big\| \mathrm{P}_\theta \left( z \right) \right). \tag{2}$$

That KL-divergence is very closely related (with a particular choice of prior, $\mathrm{P}_\theta \left( z \right)$) to the MI (Alemi et al., 2018; Chen et al., 2016b), so the KL-divergence can be understood intuitively as reducing the MI between data, $x$ and latent, $z$. However this connection between the ELBO and MI is not relevant for our work as it relates to the MI between data and latents. In contrast, our results relate to a very different quantity: the MI between different latent variables. Of course, these two MIs are entirely different quantities, so no analogies can be drawn across these approaches.

## 2.2 InfoNCE

In InfoNCE (Oord et al., 2018), there are two data items, $x$ and $x'$. Oord et al. (2018) initially describes a time-series setting where for instance $x$ is the previous datapoint and $x'$ is the current datapoint. But Oord et al. (2018) also consider other contexts where $x$ and $x'$ are different augmentations or patches of the same underlying image. We then form latent representations, $z$ and $z'$ by passing $x$ and $x'$ through neural network encoders. We consider fully general encoders encoders, $\mathrm{Q}_\phi \left( x|z \right)$ and $\mathrm{Q}_\phi \left( x'|z' \right)$ which could be stochastic or deterministic. All our derivations apply to stochastic or deterministic encoders, except Eq. (27) in Sec. 27. However, in the InfoNCE setting, we usually choose deterministic encoders,

$$\mathrm{Q}_\phi \left( z \, |x \, \right) = \delta \left( z \, - g_\phi(x) \right), \tag{3a}$$

$$\mathrm{Q}_\phi \left( z'|x' \right) = \delta \left( z' - g'_\phi(x') \right), \tag{3b}$$

which give a point distribution at the output of a neural network, $g_\phi(x)$ and $g'_\phi(x')$, where $\phi$ gives the neural network weights. Additionally, we can but do not have to choose $g_\phi = g'_\phi$. The InfoNCE objective was originally motivated as maximizing the mutual information between latent representations,

$$\mathrm{I}(\phi) = \mathrm{E}_{\mathrm{Q}_\phi(z, z')} \left[ \log \frac{\mathrm{Q}_\phi \left( z'|z \right)}{\mathrm{Q}_\phi \left( z' \right)} \right]. \tag{4}$$

Here, we are using Q rather than P for consistency with VAE derivations in the methods. The distributions, $\mathrm{Q}_\phi \left( z'|z \right)$ and $\mathrm{Q}_\phi \left( z' \right)$, are a conditional and marginal of the joint distribution, $\mathrm{Q}_\phi \left( z, z' \right)$. This joint distribution is formed by taking datapoints, $(x, x')$ drawn from the true data distribution, $\mathrm{P}_{\mathrm{true}} \left( x, x' \right)$, and encoding them with the encoders $\mathrm{Q}_\phi \left( x|z \right)$ and $\mathrm{Q}_\phi \left( x'|z' \right)$,

$$\mathrm{Q}_\phi \left( z, z' \right) = \int dx \, dx' \, \mathrm{Q}_\phi \left( z|x \right) \mathrm{Q}_\phi \left( z'|x' \right) \mathrm{P}_{\mathrm{true}} \left( x, x' \right). \tag{5}$$

Of course, $x$ and $x'$ exhibit dependencies under the true distribution, $\mathrm{P}_{\mathrm{true}} \left( x, x' \right)$, as they are e.g. the previous and current datapoints in a timeseries, or different augmentations of the same underlying image. Thus, $z$ and $z'$ must also exhibit dependencies under $\mathrm{Q}_\phi \left( z, z' \right)$. As the mutual information is difficult to compute directly, InfoNCE uses a bound, $\mathcal{I}_N(\theta, \phi)$, based on a classifier that uses $f_\theta$ with parameters $\theta$ to distinguish positive samples (i.e. the $z'$ paired with the corresponding $z$) from negative samples (i.e. $z'_j$ drawn from the marginal distribution and unrelated to $z$ or to the underlying data; see Poole et al., 2019 for further details),

$$\mathrm{I}(\phi) \geq \mathcal{I}_N(\theta, \phi) = \mathrm{E} \left[ \log \frac{f_\theta(z, z')}{f_\theta(z, z') + \sum_{j=1}^N f_\theta(z, z'_j)} \right] + \log N. \tag{6}$$

Here, the expectation is taken over $\mathrm{Q}_\phi \left( z, z' \right) \prod_j \mathrm{Q}_\phi \left( z'_j \right)$, and we use this objective to optimize $\theta$ (the parameters of $f_\theta$) and $\phi$ (the parameters of the encoder). There are two source of slack in this bound, arising from finite $N$ and a restrictive choice of $f$. To start, we can reduce but not in general eliminate slack by taking the limit as $N$ goes to infinity, (Oord et al., 2018),

$$\mathrm{I}(\phi) > \mathcal{I}_\infty(\theta, \phi) \geq \mathcal{I}_N(\theta, \phi). \tag{7}$$

The bound only becomes tight if we additionally optimize an arbitrarily flexible $f$ (Oord et al., 2018). If as usual, we have a restrictive parametric family for $f$, then the bound does not become tight (Oord et al., 2018). In reality InfoNCE does indeed use a highly restrictive class of function for $f$, which can be expected to give a loose bound on the MI (Oord et al., 2018),

$$f_\theta(z, z') = \exp\left(z^T \theta z'\right),\tag{8}$$

where $\theta$ for this particular function is a matrix. This raises the question of why we do not use a more flexible $f_\theta$ if our goal really is to maximize the MI. The answer that our goal is not ultimately to maximize the MI. Our goal is ultimately to learn a good representation, and MI is merely a means to that end. Further, Tschannen et al. (2019) argue that optimizing the true MI is likely to lead give poor repesentations, as the MI is invariant to arbitrary invertible transformations that can entangle the representation. They go on to argue that it is precisely the restrictive family of functions, corresponding to a loose bound on the MI, that encourages good representations. Tschannen et al. (2019) thus raise an important question: does it really make sense to motivate an objective that works (the InfoNCE objective) as a loose bound on an objective that does not work (the mutual information). We offer an alternative motivation by showing that the InfoNCE objective is equal to the log-Bayesian model evidence under a particular choice of prior and with a deterministic encoder.

## 3    Theoretical results

We begin by looking at the unstructured CSSVAEs with a single latent and observed variable. This gives useful intuition but does not recover InfoNCE. We then go on to look at the structured CSSVAE with two latent and two observed variables which does recover InfoNCE.

### 3.1    Unstructured CSSVAEs

In a standard variational autoencoder, we specify parametric forms (e.g. using neural networks) for the prior, $P_\theta(z)$, the likelihood, $P_\theta(x|z)$ and the approximate posterior, $Q_\phi(z|x)$. However, in an CSSVAE, we specify only the parametric form for the prior, $P_{\theta,\phi}(z)$, and the approximate posterior, $Q_\phi(z|x)$. For instance, $\phi$ might be weights of a neural network parameterising $Q_\phi(z|x)$, while $\theta$ might be the mean of a Gaussian prior. In the CSSVAE, $\phi$ are parameters that are shared across the prior and approximate posterior, while $\theta$ are parameters that are exclusive to the prior. This sharing is allowed in the variational framework: there is nothing that rules out arbitrary sharing of parameters across the generative model and approximate posterior, and indeed, this is a trick that is often used in theory and practice (Zhao et al., 2018; Ustyuzhaninov et al., 2020; Ober & Aitchison, 2021b; Aitchison et al., 2021; Ober & Aitchison, 2021a). The unusual bit is that the likelihood, $P_\phi(x|z)$, is given implicitly in terms of the approximate posterior, which has parameters $\phi$. The likelihood, $P_\phi(x|z)$, therefore depends only on the shared parameters, $\phi$, and not on the prior parameters, $\theta$. In particular, in a simple model with one latent variable, $z$, and one observation, $x$, the likelihood is given by Bayes theorem,

$$P_\phi(x|z) = \frac{Q_\phi(z|x) P_{\text{true}}(x)}{Q_\phi(z)}.\tag{9}$$

Here, $P_{\text{true}}(x)$ is the true distribution over data, which is fixed and independent of the parameters, $\theta$ and $\phi$. Of course, we cannot evaluate $P_{\text{true}}(x)$, and hence we cannot evaluate the likelihood $P_\phi(x|z)$ (it will turn out that we do not need to). Next, $Q_\phi(z|x)$ is the variational approximate posterior, parameterised e.g. as a neural network. Finally, $Q_\phi(z)$, is the normalizing constant,

$$Q_\phi(z) = \int dx\, Q_\phi(z|x) P_{\text{true}}(x),\tag{10}$$

which can be understood as the distribution over $z$ obtained by sampling $x$ from the true data distribution, and encoding using $Q_\phi(z|x)$.

Critically, $P_\phi(x|z)$ defined in Eq. (9) is a valid distribution over $x$ (albeit one whose probability density is intractable) because it is non-negative and integrates to 1. In particular, integrating, and substituting Eq. (10) into Eq. (9),

$$\int dx \, P_\phi(x|z) = \frac{\int dx \, Q_\phi(z|x) \, P_{\text{true}}(x)}{\int dx' \, Q_\phi(z|x') \, P_{\text{true}}(x')} = 1. \tag{11}$$

The model's joint distribution over $x$ and $z$ is thus,

$$P_{\theta,\phi}(x,z) = P_\phi(x|z) \, P_{\theta,\phi}(z) = Q_\phi(z|x) \, P_{\text{true}}(x) \frac{P_{\theta,\phi}(z)}{Q_\phi(z)}. \tag{12}$$

where, remember, $Q_\phi(z|x)$ is our neural network encoder, $P_{\text{true}}(x)$ is the true data distribution, $P_{\theta,\phi}(z)$ is our choice of prior, and $Q_\phi(z)$ is given by Eq. (10). Note that the model's distribution over the data can be obtained by marginalising over $z$ in Eq. 12, and we only get back the true data distribution if $P_{\theta,\phi}(z) = Q_\phi(z)$. Substituting the likelihood (Eq. 9) into the ELBO (Eq. 1), we get,

$$\mathcal{L}(x;\theta,\phi) = \log P_{\text{true}}(x) + E_{Q_\phi(z|x)}\left[\log \frac{P_{\theta,\phi}(z)}{Q_\phi(z)}\right] \tag{13}$$

where $P_{\theta,\phi}(z)$ is our parametric form for the prior and $Q_\phi(z)$ is given by Eq. (10).

Remember that $\log P_{\text{true}}(x)$ is constant with respect to the parameters as $P_{\text{true}}$ is the true, fixed data distribution. This term can thus be treated as a constant for the purposes of optimizing $\theta$ and $\phi$. Thus, to optimize $\mathcal{L}(x;\theta,\phi)$, we need to focus only on the density ratio, $P_{\theta,\phi}(z)/Q_\phi(z)$. However, this density ratio cannot be evaluated directly as we cannot evaluate $Q_\phi(z)$ (Eq. 10). Instead, inspired by InfoNCE and NCE in general, we could estimate this ratio using a classifier that distinguishes samples of $P_{\theta,\phi}(z)$ from those of $Q_\phi(z)$.

However, it turns out that this approach is unlikely to be useful for forming latent representations. In particular, consider taking the expectation of the ELBO (Eq. 13) over the true data distribution $P_{\text{true}}(x)$,

$$\mathcal{L}(\theta,\phi) = E_{P_{\text{true}}(x)}\left[\mathcal{L}(x;\theta,\phi)\right] = E_{P_{\text{true}}(x)}\left[\log P_{\text{true}}(x)\right] + E_{Q_\phi(z)}\left[\log \frac{P_{\theta,\phi}(z)}{Q_\phi(z)}\right]$$

$$= \text{const} - D_{\text{KL}}\left(Q_\phi(z)\big\|P_{\theta,\phi}(z)\right). \tag{14}$$

Optimizing the ELBO thus matches the marginal distributions in latent space between $Q_\phi(z)$ (Eq. 10) and our parametric prior, $P_{\theta,\phi}(z)$. In essence all we are doing is to find an encoder, $Q_\phi(z|x)$, from $x$ to $z$ such that, averaging over $x$ from the data, the resulting $z$'s have a distribution close to $P_{\theta,\phi}(z)$. It is not at all clear that this will give us a good representation. For instance, if $P_{\theta,\phi}(z)$ is Gaussian, and if noise in the data, $x$, is Gaussian, then it may be easier to get Gaussian $z$'s by extracting noise, rather than (as we would like), extracting high-level structure. That said, it may still be possible to do something useful by applying identifiability results inspired by ICA (e.g. Khemakhem et al., 2020).

### 3.2  Structured CSSVAEs

The previous section argued that an CSSVAE with just one latent and observed variable is unlikely to give useful representations. Instead, consider a generative model with two observed variables, $x$ and $x'$, and two latent variables, $z$ and $z'$. The approximate posterior is given in terms of neural network encoders for $x$ and $x'$ separately,

$$Q_\phi(z,z'|x,x') = Q_\phi(z|x) \, Q_\phi(z'|x'). \tag{15}$$

Remember that the InfoNCE is originally described in a time-series setting (Oord et al., 2018), where $x$ is the previous observation (or the history of past observations) and $x'$ is the current observation. Thus, we consider a Hidden Markov model like structure, where the previous and current observations, $x$ and $x'$, are associated with a latent variable at the previous, $z$, and current, $z'$, timestep. Of course, $z$ and $z'$,

are correlated, so the graphical model can be written as, $x \leftarrow z - z' \rightarrow x'$. Equivalently the generative probability can be factorised as,

$$P_{\theta,\phi}(x, x', z', z) = P_{\phi}(x|z) P_{\phi}(x'|z') P_{\theta,\phi}(z, z'). \tag{16}$$

Here, $P_{\theta,\phi}(z, z')$ should encode dependencies between $z$ and $z'$ and may be a specific, parametric form such as a Gaussian. The decoders, $P_{\phi}(x|z)$ and $P_{\phi}(x'|z')$ are given implicitly in terms of the encoders, $Q_{\phi}(z|x)$ and $Q_{\phi}(z'|x')$ and the true marginal distributions of the data, $P_{\text{true}}(x)$ and $P_{\text{true}}(x')$,

$$P_{\phi}(x|z) = \frac{Q_{\phi}(z|x) P_{\text{true}}(x)}{Q_{\phi}(z)}, \tag{17a}$$

$$P_{\phi}(x'|z') = \frac{Q_{\phi}(z'|x') P_{\text{true}}(x')}{Q_{\phi}(z')}. \tag{17b}$$

The normalizing constants, $Q_{\phi}(z)$ and $Q_{\phi}(z')$, are,

$$Q_{\phi}(z) = \int dx \, Q_{\phi}(z|x) P_{\text{true}}(x), \tag{18a}$$

$$Q_{\phi}(z') = \int dx' \, Q_{\phi}(z'|x') P_{\text{true}}(x'). \tag{18b}$$

Again, they are the distributions over $z$ and $z'$ we get by encoding samples from the true data distribution. As such, they can also be understood as marginals of $Q_{\phi}(z, z')$ in Eq. (5). Now, we follow the usual derivation of the ELBO. In particular, we start by writing the model evidence or marginal likelihood, $\log P_{\theta,\phi}(x, x')$, in terms of an integral over $z$ and $z'$ in the full generative model,

$$\log P_{\theta,\phi}(x, x') = \log \int dz \, dz' \, P_{\theta,\phi}(x, x', z, z'). \tag{19}$$

We then multiply and divide by the approximate posterior, $Q_{\phi}(z, z'|x, x')$,

$$\log P_{\theta,\phi}(x, x') = \log \int dz \, dz' \, Q_{\phi}(z, z'|x, x') \frac{P_{\theta,\phi}(x, x', z, z')}{Q_{\phi}(z, z'|x, x')}. \tag{20}$$

And rewrite the integral as an expectation under the approximate posterior,

$$\log P_{\theta,\phi}(x, x') = \log E_{Q_{\phi}(z,z'|x,x')} \left[ \frac{P_{\theta,\phi}(x, x', z, z')}{Q_{\phi}(z, z'|x, x')} \right]. \tag{21}$$

Substituting for the approximate posterior (Eq. 15) and prior (Eq. 16),

$$\log P_{\theta,\phi}(x, x') = \log E_{Q_{\phi}(z,z'|x,x')} \left[ \frac{P_{\phi}(x|z) P_{\phi}(x'|z')}{Q_{\phi}(z|x) Q_{\phi}(z'|x')} P_{\theta,\phi}(z, z') \right]. \tag{22}$$

Substituting Eq. (17) and remembering that $\log P_{\text{true}}(x)$ and $\log P_{\text{true}}(x')$ are parameter-independent constants

$$\log P_{\theta,\phi}(x, x') = \log E_{Q_{\phi}(z,z'|x,x')} \left[ \frac{P_{\theta,\phi}(z, z')}{Q_{\phi}(z) Q_{\phi}(z')} \right] + \text{const} \tag{23}$$

Finally, applying Jensen's inequality we get the ELBO for a datapoint,

$$\log P_{\theta,\phi}(x, x') \geq \mathcal{L}_{\theta,\phi}(x', x) \tag{24}$$

$$\mathcal{L}_{\theta,\phi}(x', x) = E_{Q_{\phi}(z,z'|x,x')} \left[ \log \frac{P_{\theta,\phi}(z, z')}{Q_{\phi}(z) Q_{\phi}(z')} \right] + \text{const} \tag{25}$$

And averaging over datapoints,

$$\mathcal{L}(\theta, \phi) = E_{P_{\text{true}}(x,x')} [\mathcal{L}_{\theta,\phi}(x, x')] = E_{Q_{\phi}(z,z')} \left[ \log \frac{P_{\theta,\phi}(z, z')}{Q_{\phi}(z) Q_{\phi}(z')} \right] + \text{const}. \tag{26}$$

### 3.3 Under a deterministic encoder, the CSSVAE ELBO is equal to the log-Bayesian model evidence

Remember that all our previous derivations apply to stochastic or deterministic encoders. However, we do get one additional important result for deterministic encoders (Eq. 3), namely that the ELBO is equal to the model evidence. In particular, as the encoder is deterministic, the expectations for the log-marginal likelihood (Eq. 23) and the ELBO (Eq. 25) can be evaluated straightforwardly, and are equal,

$$\log \mathrm{P}_{\theta,\phi}(x, x') = \log \frac{\mathrm{P}_{\theta,\phi}(z, z')}{\mathrm{Q}_{\phi}(z) \mathrm{Q}_{\phi}(z')} + \text{const} = \mathcal{L}_{\theta,\phi}(x', x). \tag{27}$$

so, the ELBO becomes *equal* to the log-Bayesian model evidence. This is expected if we remember that the variational bound in Eq. (25) arose from applying Jensen's inequality to Eq. (23). Importantly, looseness in the Jensen bound arises from variance in the estimator inside the expectation, $\mathrm{P}_{\theta}(z, z') / \mathrm{Q}_{\phi}(z) \mathrm{Q}_{\phi}(z')$. With a deterministic encoder, this quantity is fixed and has zero variance, implying that Jensen's bound will be tight.

### 3.4 The CSSVAE ELBO can be written as the mutual information plus a KL divergence

To get an intuitive understanding of the ELBO we take Eq. (26) and add and subtract $\mathrm{E}_{\mathrm{Q}_{\phi}(z,z')} \left[ \log \mathrm{Q}_{\phi}(z, z') \right]$,

$$\mathcal{L}(\theta, \phi) = \mathrm{E}_{\mathrm{Q}_{\phi}(z,z')} \left[ \log \frac{\mathrm{Q}_{\phi}(z, z')}{\mathrm{Q}_{\phi}(z) \mathrm{Q}_{\phi}(z')} \right] + \mathrm{E}_{\mathrm{Q}_{\phi}(z,z')} \left[ \log \frac{\mathrm{P}_{\theta,\phi}(z, z')}{\mathrm{Q}_{\phi}(z, z')} \right] + \text{const}. \tag{28}$$

The first term, is the mutual information between $z$ and $z'$ under $\mathrm{Q}_{\phi}(z, z')$ (Eq. 5), and the second term is a KL-divergence,

$$\mathcal{L}(\theta, \phi) = \mathrm{I}(\phi) - \mathrm{D}_{\mathrm{KL}} \left( \mathrm{Q}_{\phi}(z, z') \middle\| \mathrm{P}_{\theta,\phi}(z, z') \right) + \text{const}. \tag{29}$$

This objective therefore encourages large mutual information (Eq. 5), while encouraging $\mathrm{Q}_{\phi}(z, z')$ to lie close to the prior, $\mathrm{P}_{\theta,\phi}(z, z')$.

### 3.5 Under the optimal prior, the CSSVAE ELBO is equal to the mutual-information (up to a constant)

Looking at Eq. (29), the only term that depends on the prior, $\mathrm{P}_{\theta,\phi}(z', z)$, is the negative KL-divergence. As such, maximizing $\mathcal{L}$ with respect to the parameters of $\mathrm{P}_{\theta,\phi}(z', z)$ is equivalent to minimizing $\mathrm{D}_{\mathrm{KL}} \left( \mathrm{Q}_{\phi}(z, z') \middle\| \mathrm{P}_{\theta,\phi}(z, z') \right)$. Of course, the minimal KL-divergence of zero is obtained when,

$$\mathrm{P}_{\phi}^{*}(z, z') = \mathrm{Q}_{\phi}(z, z'). \tag{30}$$

For this optimal prior, the KL-divergence is zero, so the ELBO reduces to just the mutual information between $z$ and $z'$ (and a constant),

$$\mathcal{L}_{\mathrm{MI}}(\phi) = \mathrm{I}(\phi) + \text{const}. \tag{31}$$

### 3.6 Under a particular prior, the CSSVAE ELBO is equal to the infinite-sample InfoNCE objective (up to a constant)

Recent work has argued that the good representation arising from InfoNCE cannot be from maximizing mutual information alone, because the mutual information is invariant under arbitrary invertible transformations (Tschannen et al., 2019; Li et al., 2021). Instead, the good properties must arise somehow out of the fact that the InfoNCE objective forms only a loose bound on the true MI, even in the infinite sample limit Eq. (7). In contrast, here we show that the infinite-sample InfoNCE objective is equal to the ELBO (or log-Bayesian model evidence for deterministic encoders) for a specific choice of prior. In particular, we choose the prior on $z$ implicitly, as $\mathrm{Q}_{\phi}(z)$, and we choose the distribution over $z'$ conditioned on $z$ to be

given by an energy based model that depends on $Q_\phi(z')$ and an unrestricted coupling function, $f_\theta(z, z')$ (we could of course use Eq. 8 for $f_\theta$),

$$P_{\theta,\phi}^{\text{InfoNCE}}(z) = Q_\phi(z) \tag{32a}$$

$$P_{\theta,\phi}^{\text{InfoNCE}}(z'|z) = \frac{1}{Z_{\theta,\phi}(z)} Q_\phi(z') f_\theta(z, z'). \tag{32b}$$

The normalizing constant, $Z_{\theta,\phi}(z)$, is

$$Z_{\theta,\phi}(z) = \int dz' \, Q_\phi(z') f_\theta(z, z') = E_{Q_\phi(z')}[f_\theta(z, z')]. \tag{33}$$

Substituting these choices into Eq. (26), the average ELBO or log-Bayesian model evidence becomes,

$$\mathcal{L}_{\text{InfoNCE}}(\theta, \phi) = E_{Q_\phi(z,z')}\left[\log \frac{Q_\phi(z) \frac{1}{Z_{\theta,\phi}(z)} Q_\phi(z') f_\theta(z, z')}{Q_\phi(z) Q_\phi(z')}\right] + \text{const.} \tag{34}$$

Cancelling $Q_\phi(z) Q_\phi(z')$,

$$\mathcal{L}_{\text{InfoNCE}}(\theta, \phi) = E_{Q_\phi(z,z')}[\log f_\theta(z, z') - \log Z_{\theta,\phi}(z)] + \text{const}, \tag{35}$$

and substituting for $Z_{\theta,\phi}(z)$ gives,

$$\mathcal{L}_{\text{InfoNCE}}(\theta, \phi) = E_{Q_\phi(z,z')}[\log f_\theta(z, z')] - E_{Q_\phi(z)}\left[\log E_{Q_\phi(z')}[f_\theta(z, z')]\right] + \text{const.} \tag{36}$$

Following Wang & Isola (2020) and Li et al. (2021) the right hand side can be identified as the infinite sample InfoNCE objective that we introduced in Sec. 2.2,

$$\mathcal{L}_{\text{InfoNCE}}(\theta, \phi) = \mathcal{I}_\infty(\theta, \phi) + \text{const.} \tag{37}$$

Thus in this choice of model, the ELBO (or the log-Bayesian model evidence for a deterministic encoder), $\mathcal{L}_{\text{InfoNCE}}(\theta, \phi)$, is equal to the infinite-sample InfoNCE objective, $\mathcal{I}_\infty(\theta, \phi)$, up to a constant. This would argue that the InfoNCE objective has a closer link to the log-Bayesian model evidence than it does to the MI, as the infinite-sample InfoNCE objective, $\mathcal{I}_\infty(\theta, \phi)$ is exactly equal to the log-Bayesian model evidence, but in general forms only a bound on the MI,

$$I(\phi) > \mathcal{I}_\infty(\theta, \phi) = \mathcal{L}_{\text{InfoNCE}}(\theta, \phi) + \text{const.} \tag{38}$$

Of course, in practice, InfoNCE uses finite samples, because the infinite limit in Eq. (36) is intractable. Likewise, if we were to practically use a CSSVAE with this prior, we would also be forced to use finite samples. The solution in both cases would be exactly the same, to use the bound given by the usual finite-sample estimator, as originally described in the InfoNCE framework (Oord et al., 2018).

## 4 Experimental results

Our primary results are theoretical: in connecting the log-Bayesian model evidence and mutual information, and in showing that the InfoNCE objective with a restricted choice of $f_\theta$ makes more sense as a bound on the log-Bayesian model evidence than on the MI. At the same time, our approach encourages a different way of thinking about how to set up contrastive SSL methods, in terms of Bayesian priors. As an example, we considered a task in which the goal was to extract the locations of three moving balls, based on videos of these balls bouncing around in a square (Fig. 1A; Appendix A).

In particular, we apply the InfoNCE-like setup described in Sec. 3.6. Our prior is given by Eq. (32), with $Q_\phi(z)$ and $Q_\phi(z')$ defined by Eq. (18). The freedom in this setup is given by the choice of $f_\theta$. Naively applying the usual InfoNCE choice of $f_\theta$ (using Eq. 8), failed (linear in Fig. 1BC), because we did not correctly encode prior information about the structure of the problem. Critically, our prior is that for the

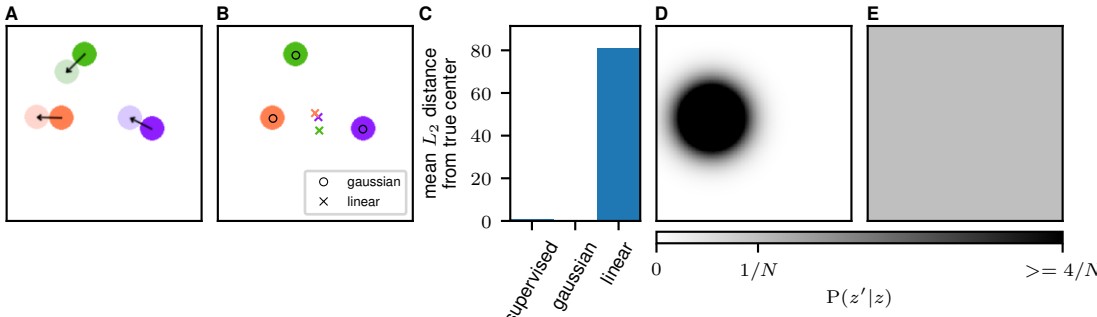

Figure 1: Results of the moving balls experiment. **A**) Example of the motion between consecutive frames. The balls move by a full diameter in a semi-random direction. **B**) Locations of the extracted ball centres, after supervised linear decoding. The standard InfoNCE setup fails to extract correct locations. **C**) The mean distance from the extracted and true centres of the balls for a supervised method, InfoNCE with a Gaussian discriminator after supervised decoding and InfoNCE with a linear discriminator after supervised decoding. **D**) Probability distribution for the next location of the coral ball in **A** according to an encoder trained with a Gaussian discriminator. **E**) Probability distribution for the next location of the same ball according to an encoder trained with a linear discriminator.

adjacent frames, the locations extracted by the network will be close, while for random frames, the locations extracted by the network will be far apart. The linear estimator in Eq. (8) is not suitable for extracting the proximity of the ball locations, so it fails (linear in Fig. 1 BC). In particular, it corresponds to a non-sensical prior over $z'$ given $z$,

$$\mathrm{P}_{\theta,\phi}^{\mathrm{InfoNCE}}\left(z'|z\right) = \frac{1}{Z_{\theta,\phi}(z)}\, \mathrm{Q}_\phi\left(z'\right) f_\theta(z,z') \propto \exp\left(z^T\theta z'\right) \tag{39}$$

(where we have taken $\mathrm{Q}_\phi\left(z'\right)$ defined by Eq. 18 to be approximately uniform purely for the purposes of building intuition). This prior will encourage $z^T\theta z'$ to be as large as possible, which could be achieved for instance by setting $z' = \lambda\theta z$ with very large $\lambda$. Instead, we would like a prior that encodes our knowledge that $z'$ is likely to be close to $z$. We can get such a prior by using a Gaussian RBF form for $f_\theta$,

$$f_\theta(z,z') = \exp\left(-\tfrac{1}{2\theta^2}(z-z')^2\right). \tag{40}$$

where the only parameter, $\theta$, is a scalar learned lengthscale. Critically, this choice of $f_\theta$ is natural and obvious if we take a probabilistic generative view of the problem (with a uniform $\mathrm{Q}_\phi\left(z'\right)$, this corresponds to a Gaussian conditional, $\mathrm{P}_{\theta,\phi}^{\mathrm{InfoNCE}}\left(z'|z\right)$). In contrast, if our goal was to maximize information, then the most appropriate choice would be an arbitrarily flexible $f_\theta$.

## 5 Related work

Perhaps the closest prior work is Zimmermann et al. (2021), which also identifies an interpretation of InfoNCE as inference in a principled generative model. Unlike this work, we identify a connection between the InfoNCE objective and the ELBO or model evidence. In addition, their approach requires four restrictive assumptions. First, they assume deterministic encoder, e.g $\mathrm{Q}_\phi\left(z|x\right) = \delta\left(z - g_\phi(x)\right)$. In contrast, all our theory applies to stochastic encoders. While we do explicitly consider deterministic encoders in Sec. 3.3, this is only to show that with deterministic encoders, the ELBO bound is tight — all the derivations outside of this very small section (which includes all our key derivations) use fully general encoders, $\mathrm{Q}_\phi\left(z|x\right)$ and $\mathrm{Q}_\phi\left(z'|x'\right)$. Second, they assume that $z_\phi(x)$ is invertible, i.e. that there exists a deterministic decoder $x_\phi(z)$, which is not necessary in our framework. This is a particularly problematic assumption as practical encoders commonly used in contrastive SSL are not invertible. Third, they assume that the latent space is unit hypersphere, while in our framework there is no constraint on the latent space. Fourth, they assume the ground truth

marginal of the latents of the generative process is uniform, whereas our framework accepts any choice of ground-truth marginal. As such, our framework has considerably more flexibility to include rich priors on complex, structured latent spaces.

Other work looked at the specific case of isolating content from style (von Kügelgen et al., 2021). This work used a similar derivation to that in Zimmermann et al. (2021) with slightly different assumptions. While they still required deterministic, invertible encoders, they relax e.g. uniformity in the latent space. But because they are working in the specific case of style and content variables, they make a number of additional assumptions on those variables. Importantly, they again do not connect the InfoNCE objective with the ELBO or model evidence.

Very different methods use noise-contrastive methods to update a VAE prior (Aneja et al., 2020). Importantly, they still use an explicit decoder.

There is a large class of work that seeks to use VAEs to extract useful, disentangled representations (e.g. Burgess et al., 2018; Chen et al., 2018; Kim & Mnih, 2018; Mathieu et al., 2019; Joy et al., 2020). Again, this work differs from our work in that it uses explicit decoders and thus does not identify an explicit link to self-supervised learning.

Likewise, there is work on using GANs to learn interpretable latent spaces (e.g. Chen et al., 2016a). Importantly, GANs learn a decoder (mapping from the random latent space to the data domain). Moreover, GANs use a classifier to estimate a density ratio. However, GANs estimate this density ratio for the data, $x$ and $x'$, whereas InfoNCE, like the methods described here, uses a classifier to estimate a density ratio on the latent space, $z$ and $z'$.

There is work on reinterpreting classifiers as energy-based probabilistic generative models (e.g. Grathwohl et al., 2019), which is related if we view SSL methods as being analogous to a classifier. Our work is very different, if for no other reason than because it is not possible to sample data from an CSSVAE (even using a method like MCMC), because the decoder is written in terms of the unknown true data distribution.

## 6 Conclusions

In conclusion, we have developed a new family of contrastive VAE, CSSVAEs. The CSSVAE ELBO is equal to the log-Bayesian model evidence for a deterministic encoder. For the optimal prior, the CSSVAE ELBO is equal to the mutual information, and with a particular choice of prior, the CSSVAE ELBO is equal to the infinite-sample InfoNCE objective (up to constants). In contrast, the infinite-sample InfoNCE forms only a loose bound on the true MI, which would argue that the InfoNCE objective might be better motivated as the CSSVAE ELBO. As such, we unify contrastive semi-supervised learning with generative self-supervised learning (or unsupervised learning). Finally, we provide a principled framework for using simple parametric models in the latent space to enforce disentangled representations, and our framework allows us to use Bayesian intuition to form richer priors on the latent space.

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

## A Experimental details

We generated 900 images in a single continuous video with a resolution of $256 \times 256$ pixels. The three balls had a diameter of 32 pixels. Between consecutive frames the balls moved by a full diameter in a random direction, as illustrated in Fig. 1A. The movement trajectory was picked by taking the previous trajectory and adding a uniform noise of $-2°$ to $+2°$. If the picked movement resulted in a collision, we sampled a new trajectory by doubling the noise range until a valid trajectory is found.

We trained the model in a classic self-supervised manner. We encoded one "base" frame, one "target" frame (the next frame in a video sequence), along with a number of random frames. As usual, the network was trained to distinguish between the target frame (adjacent to the base frame) and random frames. We then trained a linear decoder in a supervised manner to return the $(x, y)$ locations of the balls.

The encoder itself is a simple convolutional neural network, as shown in Fig. 2. It consists of 2 batch normalised convolutional layers with a kernel size of 3. The first layer uses ReLU as the activation function, while the second layer uses a sigmoid. At the output of the convolutional layers, we have 3 feature maps, which we interpret as the locations of the 3 different balls. We finally extract these locations by computing the centre of mass of the feature maps, giving a vector of six numbers as output (the x and y locations of the centres of mass of each feature map). The training itself was performed by using stochastic gradient descent with a learning rate of 0.005 over the course of 30 epochs. The batches were made of 30 random pairs of consecutive frames. For any pair, we use the second frame as the positive example and we use the second frame of the other pairs in the batch, as the random negative examples, against which we contrast.

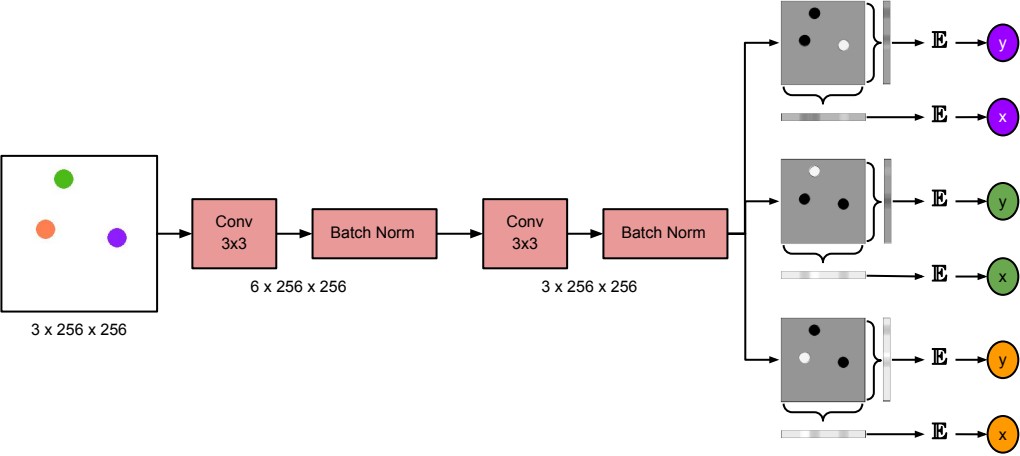

Figure 2: Architecture of the encoder neural network. The first of the two 3x3 convolutional layers outputs 6 feature maps and uses a ReLU activation. The second convolution outputs 3 feature maps and applies a sigmoid activation. For each of these 3 maps, we extract their centre of mass. This is done by summing each dimension and normalising it to 1. This is then used to perform a weighted average over the axis locations and get the final coordinates.

