# OpenReview forum: "InfoNCE is a variational autoencoder"
_TMLR — Rejected by TMLR_

### Review · Reviewer_ZXW3 · 2022-10-20

**Summary Of Contributions:**

This paper proposes an ELBO objective that can recover objectives in the InfoNCE approach. The derivation lies on a specification of the VAE model in the "reverse direction": the data distribution and a variational posterior-like encoder distribution. Further, the contrastive nature is introduced by specifying a two-variable version of the ELBO, which relies on an energy function linking the latent variables of two data points. Then, under appropriate choices of the prior on the two-variable joint latent, version of InfoNCE objectives can be established. The experiments verify that the choice of prior is important through a synthetic dataset.

**Audience:**

Yes

**Broader Impact Concerns:**

Not really applicable to this work.

**Claims And Evidence:**

Yes

**Requested Changes:**

### Critical points
Updating the paper and/or giving a satisfactory explanation of the following is critical.
1. I don't think section 3.3 is theoretically sound or relevant. With a delta function, it's not even clear if the log ratio $P(z,z')/(Q(z),Q(z'))$ is well-defined. It is also unclear how $\log P(x,x') = \mathcal{L(x,x')}$ in this case. More explanation is needed. However, this section really is quite irrelevant to the main message of the paper so the review would suggest cutting it entirely unless I'm missing something important.
2. A conceptual question: throughout the paper, there is an expectation taken over the joint data distribution between data points $P_\text{true}(x,x')$. By definition, prima facie this factorises into $P_\text{true}(x)P_\text{true}(x')$. With the independence assumption in (12), the exact prior over (Z, Z') defined through $Q$ and $P_\text{true}$ should also be independent. Thus, there is no reason to have $P(z, z')$ as it introduces a redundant dependence. However, on second look, throughout the paper, it's been assumed that (x, x') are actually correlated through spatial and temporal correlations, as they are usually taken from a bigger image or nearby segments on time series. For correctness, I would emphasize that (X, X') are actually correlated, and ideally, the expectation over data is taken under $P_\text{true}(X|Y)P_\text{true}(X|Y)P_\text{true}(Y)$, where $Y$ is e.g. the image identity that make $X$ and $X'$ marginally correlated.
6. Assuming still the prima facie definition of $P_\text{true}(X, X')$, the comparison between 3.5 and 3.6 raises an interesting concern: under the stated independence assumption (12), the prior joint $P_\text{MI}(Z,Z')=Q(Z,Z')$ is also independent, which is a more restrictive than the form in (29)-(30). But how come the former recovers the exact MI objective, while the more flexible latter gives the InfoNCE objective which is a lower bound on the MI? It feels as if a more restrictive approach turned out to give a stronger objective for maximizing mutual information. Is this a correct interpretation?
7. The authors seem to have only shown the correspondence to the infinite sample limit of the info-NCE objective. However intermediate values of N? If the results only hold in the infinite-N limit, then I wouldn't really call this a unification as stated in the abstract.
8. Further the experimental sections need more detailed explanations, ideally with equations. For example, I don't immediately see how the linear decoder is constructed and why it corresponds to the quadratic form in the exponent in (34).
9. On (34), how can we just take $Q(z)$ to be uniform? Isn't this also a nonsensical prior, as no non-trivial encoders would give you that aggregate posterior right?

### Detailed points
Addressing the following points would strengthen the paper:
1. In equation 2, it is worth defining Q(z,z') before as Q has been used for a VAE model.
2. I don't quite follow the statement 6 lines down from (6). Why are they equivalent when they share parameters? Shouldn't there by any structural/functional constraints as well before mentioning parameter sharing?
3. There is at least one recent work by [Walker et al](https://arxiv.org/pdf/2209.05661.pdf) that defines a generative model by the encoder, although the purpose is quite different. So I suggest the authors remove the (quite irrelevant) novelty statement.
6. It might be worth mentioning that choice of $P(z, z')$ in (13) is important: it should not be factorised, because a factorisation will render everything in Section 3.2 the same as Section 3.1
7. I fully understand and appreciate the disclaimer on experimental richness in Section 4. If the paper is accepted, I would strongly advise removing those statements.

**Strengths And Weaknesses:**

## Strengths:
1. The main idea of the paper is written clearly, with all derivations and assumptions explicitly stated.
2. The theoretical contribution that links VAE with InfoNCE is correct, novel (to my knowledge), interesting and significant. In particular, the analyses in 3.6 is elegant and give me a good thrill.
3. The experiment clearly makes the point that prior choice is crucial.

## Weaknesses
1. The definition of SSVAE was never clearly stated but gets references as clearly as in the abstract.
2. I have a few disagreements and questions about some technical details. See below.
3. The experimental should be extensively expanded to support the important theoretical claims and results in the analyses (not to get any performance). See below

---

> ### Author Response · Authors · 2022-11-03
> **Response**
>
> Thanks for your positive review!
>
> The reference to the SSVAE (now CSSVAE) has been dropped from the abstract, and has been defined in Sec. 3.1 and 3.2 as VAEs with a particular choice of likelihood, written in terms of the encoder.
>
> Critical points:
> 1. Section 3.3 is theoretically sound, because none of the distributions, $P(z, z')$, $Q(z)$ or $Q(z')$ is a delta-function.  For instance, $Q(z)$ is defined in Eq. 20 as $Q(z) = \int dx Q(z| x) P_{\rm true}(x)$ so $Q(z)$ is not a delta-function, even if $Q(z|x)=\delta(z-g(x))$ is a delta-function (as long as $g(x)$ varies with $x$, and isn't a constant).
>
> 2. $x$ and $x'$ are not independent under the true distribution, otherwise, we agree, not much we do would make sense.  We have added a note about this as we introduce $P_{\rm true}(x, x')$ (just under Eq. 5).
>
> 3. $Q(z, z')$ is not independent (Eq. 4) (this definition was in the original manuscript, but much later on).  We have again clarified this point.
>
> 4. In practice, neither the infinite-sample InfoNCE objective, nor the exact ELBO can be computed exactly, as they both require an infinite limit.  As the infinite-sample InfoNCE objective and exact ELBO are equivalent, it would make sense to use the same finite-sample bound, and the most obvious choice would be that from Oord et al. (2018). As such, we would in practice use exactly the finite-sample InfoNCE objective to optimize a bound on the ELBO/log-Bayesian model evidence in a CSSVAE.  We have added a note on this point to the end of that section.  However, there is perhaps a more important point that we have updated the paper to clarify.  In particular, for the InfoNCE objective to equal the true MI, we need to be in the infinite sample limit, _and_ we need to optimize an arbitrarily flexible $f$.  In contrast, for the InfoNCE objective to equal the ELBO/Bayesian model evidence, we only need the infinite sample limit.  Given that in practice, InfoNCE uses a highly restrictive family for $f$, this would argue that the InfoNCE objective would seem to be more closely related to the ELBO/Bayesian model evidence than the true MI.
>
> 5. We have been much more careful to specify exactly the choice of prior in the experiments, giving explicit references to all the relevant equations.  The linear decoder was not constructed specifically for this problem.  It just comes from the InfoNCE paper (Eq. 11).
>
> 6. $Q(z)$ is defined by Eq. 19, so we can't choose it to be anything else.  In this case, we agree that we expect $Q(z)$ to be close to but not exactly uniform.  We could of course choose $P(z)$ to be uniform, which would impose additional inductive biases.  However, in that case, we would not be able to use the slightly simpler InfoNCE like objective, and would instead need to use the full objective in Eq. 28.
>
> Detailed points:
> 1. Done (moved the definition of $Q(z, z')$ earlier).
>
> 2. We have clarified this statement.  The intention here is to claim that the family of approximate posteriors and likelihood parameterised by Eq. 19 could also be written out more explicitly in terms of parameters that are shared between the likelihood and approximate posterior in a very specifically chosen highly constrained family of distributions.  The intention was not to claim that any family of approximate posteriors and likelihoods can be written in the form of Eq. 19 (which is manifestly not true).
>
> 3. We have removed that novelty statement and added a reference to Walker et al. However, Walker et al. is very recent work that cites us, that was released on arXiv in the last month or two.  In contrast, our work has been available on arXiv for over a year.
>
> 4. We have added this note.
>
> 5. Thanks for this!  We fully agree and have implemented these changes.
>
>
> We have modified Sec. 3.3 to make this connection more explicit.  In particular, the result arises just from applying the deterministic encoder to the definition of the log-marginal likelihood (Eq. 23) and the ELBO (Eq. 25).

---

> ### Comment · Reviewer_ZXW3 · 2022-11-24
> **Representation learning**
>
> I thank the authors for the answers. I'm happy with most answers but still want to confirm some points.
>
> When substituting (12) and (13) into (16), the Q under the expectation should also change to the factorised version as in the square brackets. This doesn't affect anything substantial, is this right?
>
> One thing I think would make the paper much clearer is to make the dependence between x and x' explicit throughout the derivations. The dependence is through an original image x0 from which crops x and x' are drawn. Without making it explicit, I don't know which distribution should carry the original x0. This is rather important because what's really doing the job of representation learning is (26) where the loss is optimized across dependent pairs x and x': The first KL in (26) is crucial as that will become the mutual information across encodings of crops from the same image. At least this is my understanding; is this a reasonable understanding?

---

> > ### Author Response · Authors · 2022-11-26
> > **Response**
> >
> > > When substituting (12) and (13) into (16), the Q under the expectation should also change to the factorised version as in the square brackets. This doesn't affect anything substantial, is this right?
> >
> > I'm not 100% sure what's going on here, as the equation numbers may have shifted.  In any case, there are some expectations (e.g. Eq. 21) under $Q(z, z'| x, x')$.  Given that (Eq. 15)
> > \begin{align}
> > Q(z, z'| x, x') = Q(z| x) Q(z'|x')
> > \end{align}
> > they could equally be written as expectations under $Q(z| x) Q(z'|x')$, which wouldn't affect anything.  But this doesn't apply to Sec. 3.1 as there is only one datapoint and latent variable in that setting.
> >
> > > One thing I think would make the paper much clearer is to make the dependence between x and x' explicit throughout the derivations. The dependence is through an original image x0 from which crops x and x' are drawn. Without making it explicit, I don't know which distribution should carry the original x0. This is rather important because what's really doing the job of representation learning is (26) where the loss is optimized across dependent pairs x and x': The first KL in (26) is crucial as that will become the mutual information across encodings of crops from the same image. At least this is my understanding; is this a reasonable understanding?
> >
> > Correct.  We've reminded the reader of where these x's are coming from a couple more times (e.g. when we define the generative model).

---

### Review · Reviewer_McsS · 2022-10-25

**Summary Of Contributions:**

The paper rewrites the ELBO from variational inference in a particular way consisting of two KL divergence terms, and points out that one of these terms is the InfoNCE objective and the other is the KL divergence between a parameterized choice of prior and some other distribution which we cannot actually evaluate. They then point out that this implies that InfoNCE is optimizing this objective with an implicit prior.

**Audience:**

Yes

**Claims And Evidence:**

Yes

**Requested Changes:**

I think the biggest issue with this paper is that most of it is just rewriting terms, and it's unclear what the takeaway is.

It would go a long way to drop most of the algebraic manipulation (or move it to the appendix) and instead spend some more space clearly and precisely describing what is the story here. What is the advantage to using other priors? Is this not what people are already doing? If it's what they're doing only implicitly (which, for the record, I don't think is the case: I think some papers have done this quite explicitly), what is the benefit of this connection?

The paper doesn't have to actually present a method that *uses* this benefit, but as it stands I really don't see any way in which rewriting the ELBO in this way is useful (if even new).

**Strengths And Weaknesses:**

I've read through this paper twice now and I'm still not really clear on what point the paper is making.

The authors takes 4-5 pages and a lot of basic algebra to rewrite the ELBO in many different ways, eventually arriving on a term consisting of two KL divergences (Eq. 26). One of these terms is the mutual information, so the paper points out that if we assume an implicit prior which makes the second divergence equal to 0, optimizing this objective is the same as mutual information maximization. Alternatively, we could choose a different prior (such as one induced by an energy function f(z, z')), which would give a different objective.

All of these points have been approximately used in previous works [1, 2, 3]. I don't know if these sources have directly written out "you can write the InfoNCE objective in this way, and therefore it is a particular choice of prior", but I don't feel this exact statement gives us any new information. For example, [4, 5, 6] already consider settings where we might optimize a different prior, such as a von-Mises distribution or general convex bodies with log-bilinear distributions.

I've kept in mind that this is TMLR, and so I asked myself if I felt anyone would gain anything from reading this work, regardless of whether the ideas themselves are novel. In all honesty, I believe the answer is no. Most of this paper consists of simple algebraic manipulation and it doesn't feel like there's any real takeaway at the end---if there is, I was unable to identify it after two readthroughs. The earlier writing is reasonably clear but everything after the intro just left me in a haze.

Finally, I'm quite confused as to the choice of the name "Self-Supervised VAE". VAEs are already unsupervised. Also, the final objective which is written as the SSVAE objective does not involve any sort of reconstruction loss as far as I can tell, so is it really an autoencoder? I think really this is just "variational inference with neural networks".

[1] Contrastive Multiview Coding. Tian et al.

[2] A Simple Framework for Contrastive Learning of Visual Representations. Chen et al.

[3] Provable Guarantees for Self-Supervised Deep Learning with Spectral Contrastive Loss. HaoChen et al.

[4]  Self-supervised learning with data augmentations provably isolates content from style. von Kügelgen et al.

[5] Contrastive learning inverts the data generating process. Zimmermann et al.

[6] f-Mutual Information Contrastive Learning. Zhang et al.



================================================

Update: the authors have made several changes to the earlier writing which I think clarifies quite a bit, as well as renaming the model they consider. While I still feel this work could benefit from additional rewriting, at this point I would consider it above the bar for publication at TMLR. However, this remains subject to correctness concerns, which are not mine (see other reviewers). This is not really my area of expertise so I don't feel qualified to comment on this point.

---

> ### Author Response · Authors · 2022-11-03
> **Response**
>
> We have shifted to the terms generative and contrastive SSL, which hopefully reduces confusion.
>
> Our most prosaic contribution follows from your references 4,5.  These references and the contribution are already discussed at length in the original related work.  In particular, these papers establish that connecting generative and contrastive SSL is interesting and   has a broad audience.  However, their results require a number of strong assumptions including to, deterministic, invertible encoders.  Perhaps the restriction to deterministic encoders is okay (though we might very well be interested in stochastic encoders). But the restriction to invertible encoders is very problematic as invertible encoders are rarely used, as they requires imposing strong, non-standard structure on the network.  In contrast, our results are far more general, and apply to arbitrary, non-invertible, stochastic or deterministic encoders with no geometric restrictions on the latent space.  Importantly, we were able to obtain such general results because of our insights into non-standard construction of a VAE.  Once we have that non-standard construction, we were indeed able to show equivalence to InfoNCE by relatively straightforward algebraic manipulation (indeed, that's part of the strength of our approach!)
>
> There are at least two other more profound contributions we have updated the Abstract, Background and Results to emphasise these results.
> 1. We show that for a certain prior, and with a deterministic encoder, the mutual information is _equal_ to the Bayesian model evidence (up to a constant factor).  This is a profound connection between perhaps the most important quantity in Bayesian inference (the     Bayesian model evidence) and perhaps the most important quantity in Information theory (the mutual information), which does not appear in 1-6.
>
> 2. Even in the infinite-sample limit, the InfoNCE objective forms only a loose bound on the MI (it only becomes tight when we optimize an arbitrarily flexible $f$, while in practice we use a highly restrictive $f$).  This would seem to call into question the motivation for the InfoNCE objective as a bound on the MI.  Indeed, Tschannen et al. (2019) has called this link further into question; they note that the true MI can lead to arbitrarily entangled representations, and thus is unlikely to give good representation learning.  Tschannen et al. (2019) thus raise a key question: does it really make sense to motivate an objective that works (i.e. the InfoNCE    objective) as a loose bound on an objective that does not work (i.e. the true MI which gives arbitrarily entangled representations). We show that the infinite sample InfoNCE objective is equal to the ELBO/log-Bayesian model evidence, while only being a loose bound on the MI.  This would seem to argue that we should motivate the InfoNCE objective as the log-Bayesian model evidence (as they're equal with infinite samples), rather than as a losoe bound on the MI (as we only have a bound, even in the infinite sample case).  Again, this point does not appear in 1-6.
>
> We have renamed the method "Contrastive Self-Supervised VAEs" or CSSVAEs, to highlight their contrastive nature.  CSSVAEs are radically different from VAEs.  For instance, in a VAE, we can easily sample $x$.  However, we cannot sample $x$ in a CSSVAE as we cannot evaluate the probability density of the likelihood.  This is because a standard VAE has an explicit parameteric form for the likelihood, for instance involving a neural network.  In contrast, a CSSVAE defines the likelihood implicitly in terms of the encoder and the true data distribution, which is unknown.  Ultimately, a CSSVAE can be understood as a different interpretation of a self-supervised model using an InfoNCE-like objective, so it behaves like such a self-supervised model, and not like a classical VAE.
>
> Finally, we're slightly confused about some of the papers cited:
>
> 1+2 reinforce the motivation for our work, by highlighting that contrastive and generative SSL are considered very distinct approaches:
> > To an autoencoder, or a max likelihood generative model, a bit is a bit. No one bit is better than any other. Our conjecture in this paper is that some bits are in fact better than others.
>
> > However, pixel-level generation is computationally expensive and may not be necessary for representation learning.
>
> 3: Does not mention generative models at all (except when they talk about generating toy data upon which to test their approach).
>
> 4+5: see above.
>
> 6: This appears to be a standard InfoNCE approach, with a modified mutual information based on f-divergences.

---

> > ### Comment · Reviewer_McsS · 2022-11-24
> > **Update**
> >
> > Thanks for your response. I think the renaming (and the modified Introduction) should help clarify things.
> >
> > Your explanation here was helpful---I admit this is not exactly my area of expertise, and it's a bit odd that this review does not have an option to indicate "confidence". I'm not really the best person to assess the correctness of this work, nor its contribution (Reviewer 5HXJ seems more qualified). Ultimately, my main consideration was the clarity.
> >
> > The writing itself is reasonably clear. That is, I can understand each claim being made and how each point follows from the previous one. My bigger concern is the overall organization and focus: as I said in my original review, I really struggled to understand what exactly was the main point being made and why it matters, rather than getting bogged down by simple algebraic manipulation. Generally, I think most of these mathematical steps do not help with any sort of *intuitive understanding* of the point being made, which is the only reason that math should be included in the main body.
> >
> > I would say that conditioning on the results being correctly presented and contextualized (an assessment which I'll leave to other reviewers), I think this is probably fine for publication in TMLR. But I encourage the authors to put further effort into **very clearly** identifying the contribution of this work---the new ending paragraph of the Introduction is a major improvement over the previous version and a good example of how much of a difference clear writing can make.
> >
> > Re: the sources I provided, [3] does have a generative model. They do not *fit* that model, but they do model the underlying manifold with a metric given by a kernel over image augmentations, and this gives rise to a different "InfoNCE-like" objective.

---

### Review · Reviewer_5HXJ · 2022-11-02

**Summary Of Contributions:**

The paper makes a claim that specific forms of the variational autoencoder objective can be interpreted as mutual information maximization, and thus is naturally connected to InfoNCE. The paper performs an empirical study showing the importance of choosing the correct function form in InfoNCE in an experiment.

**Audience:**

No

**Broader Impact Concerns:**

N/A.

**Claims And Evidence:**

No

**Requested Changes:**

Given the state of the paper, unfortunately, I don't think there are much that could make me recommend acceptance. I would recommend the authors to clear some confusions regarding the joint distribution over $x, x', z, z'$ that is being discussed. I would also advise a revision over the tone of some of the claims made, for example:
> Specifying the likelihood, P(x|z), in terms of the approximate posterior, Q (z|x), is highly unusual in the variational framework — indeed, we believe we are the first to propose it.

I would gracefully disagree, since there is actually a UAI 18 work by Zhao et al. that holds the same view: The Information Autoencoding Family: A Lagrangian Perspective on Latent Variable Generative Models, see Equation (3).

**Strengths And Weaknesses:**

I will focus on the weaknesses, since I don't think the main claim in the paper is true.


**Contradictions to earlier known results**

The well-known result in Barber and Agakov states that you can maximize a lower bound of the mutual information objective simply by computing an autoencoding objective, and in other known results from Alemi (Fixing a broken ELBO) or Chen et al. (Variational Lossy Autoencoder) and mentioned in the introduction, VAE objective does not encourage higher mutual information between $x$ and $z$, due to the additional KL term (this is also related to $\beta$-VAE). So the claim here alone that VAE amounts to InfoNCE (and mutual information) already seems fishy to me.

**The claim is incorrect because of Incorrect characterizations of the graphical model that is being discussed**

Naturally, it is unfair to refute the claims only based on prior conclusions, so I dug a little deeper into the details.

- What is the "mutual information" compared over? In representation learning, the mutual information is performed between the data and the corresponding latent variable, i.e., $x$ and $z$.  If your graphical model is something like:
$z_1 \leftarrow x_1 \leftarrow x \to x_2 \to z_2$, then by data processing inequality, higher mutual information between $z_1$ and $z_2$ does indicate higher mutual information between $z_1$ and $x_1$, etc. This is the common case for explaining contrastive learning-like methods in InfoNCE, and naturally there is no need to have a decoder.
- However, the graphical model in this paper is the "generative" route, see line before Equation 13. This is important as it means that the prior $P(z, z')$ *have to be defined first* (or learned jointly in some VAEs), and $Q(z, z')$, the aggregate posterior, needs to be learned somewhat based on that (it is not the other way around, as in Eq. (27)). This means that in Eq. (26), it is unfair to simply set the second KL divergence to be zero, and just keep the first mutual information term -- you have to optimize both terms! In fact, the conclusion that can be drawn from the derivations is that "autoencoders are InfoNCE" (well, that is also in-line with the result from Barber and Agakov), because you have artificially dropped the second KL divergence, meaning that the prior is meaningless during optimization of the so-called VAE (you just set it to be the same as the aggregate posterior $q(z, z')$, so how is this different from a regular autoencoder?).
- Okay, so even though the conclusion in the paper "is so strong that experimental comparisons become meaning less", maybe we can still find some merits in the experiments? Essentially, the $f(z, z')$ function is replaced from an exponential bilinear function to an RBF function, and the authors claim that "this notion is not possible to come from the mutual information estimator viewpoint". This is clearly an over-statement: why not use the Barber and Agakov viewpoint, where you simply encode $z$ such that it is useful for predicting the next frame? Again, mutual information between $z$ and $z'$ is sometimes (under correct assumptions over the graphical model) simply a surrogate to the true thing we care about, which is the mutual information between $x$ and $z$, so we can also apply principles that work for mutual information estimation with $x$ and $z$. And the only real conclusion I can draw is that you can change the $f$ function such that it works better empirically -- does it actually give any insights regarding representation learning (say the settings in contrastive learning) in general?

Again, with the derivations, I would settle with the claim that "autoencoders are InfoNCE", but that is not exactly unknown.

---

> ### Author Response · Authors · 2022-11-03
> **Response**
>
> We suspect that the reviewer was misled by trying to understand our work in terms of two sets of prior art that are only very distantly relevant to our work.
>
> First, we agree with the reviewer and Alemi and Chen et al., that the KL term in VAEs tends to decrease (not increase) the MI between data, x, and latents, z. Of course, these results are not relevant to our work, because we do maximize the MI between data and latents.   Instead, we maximize the MI between two different latent variables. Of course, maximizing the MI between different sets of random variables will have a very different effect on the model, so intuition from Alemi and Chen et al. does not apply to our work.  Given that this might be a common confusion, we have added a paragraph on this point to the Background (Sec. 2.1).
>
> Second, we agree with the reviewer and Barber and Agakov (2003) that one can relate information maximization to an autoencoder.  We just aren't quite sure what that has to do with our work (except perhaps as a distant precursor to InfoNCE?).  Importantly, they draw this  relationship for autoencoders, not variational (Bayesian) autoencoders. And of course, variational (Bayesian) autoencoders are very, very different from autoencoders (VAEs are much more closely related to Variational Bayes, and bear only superficial similarities to auto encoders).  Moreover, we have a structured model, with two observations and two latents, while they simply have the input and output of a channel.
>
> Understanding our work in terms of this prior art is difficult, precisely because our work is so different. However, even given these issues, there appears to be no disputes around almost all of our key conclusions. The most difficult step --- of writing the likelihood in terms of the encoder and true data distribution is regarded as unproblematic. Once we accept that point, everything else follows straightforwardly by merely "shuffling symbols" (from reviewer McsS). Most importantly, our key result that with a particular choice of prior, the ELBO/log-Bayesian model evidence recovers the actual InfoNCE objective (which is related to, but is not the MI) is again not disputed.
>
> Bullet 1: See above for MIs.  Of course we are aware of the DPI.  And of course in InfoNCE, there is no need for a decoder (nor is there any need to actually evaluate the decoder in a CSSVAE).
>
> Bullet 2: We have revised the manuscript to highlight that this prior is not (necessarily) a choice.  It emerges naturally if we optimise the ELBO wrt the parameters of a sufficiently flexible prior, $P(z, z')$ with a fixed encoder.  In any case, this point is relevant only to Sec. 3.5. For the key results on connections with the InfoNCE objective in Sec. 3.6, we make a different choice of prior.
>
> We have removed the claim of priority in response to another reviewer.  But it was not at all obvious to us how Zhao et al. (2018) Eq. 3 implies that they're using a likelihood that is parameterised in terms of the encoder distribution (as in our Eq. 11).  In arXiv:1806. 06514v2 (just in case we're looking at different Eq. 3 in different versions), Eq. 3 is just $\max_\theta -D_{KL}(q_\theta(x,z)||p_\theta(x,z))$. Of course, the encoder and decoder here do share parameters, $\theta$, but we already discuss several prior works that share parameters to good effect.  We have therefore added this citation to that list.

---

> > ### Comment · Reviewer_5HXJ · 2022-11-18
> > **Can you choose a Bayesian prior based on all the training data?**
> >
> > I appreciate the authors time and effort to respond and update the paper.
> >
> > Unfortunately, I am not convinced by the updated result, and I still believe the claims are wrong (or at the very least, confusing or misleading). I believe that the math is not wrong, but the interpretation of the math is. The key issue is that $P(z, z')$ becomes data-dependent, and is not a Bayesian prior (because it takes evidence into account), making the framework more akin to justifying autoencoders than variational autoencoders. Similarly for InfoNCE derivation, (32) indicates that the "prior" depends on the function $f$ that is to be optimized over data.
> >
> > In the new draft, Section 3.5, (30), the "prior" is optimized such that $P(z, z') = Q(z, z')$; this needs to happen in order for us to have (31) which is the mutual information term, and it suggests that "prior" is a something that you can optimize. In Section 3.1, it is said that the prior and the approximate posterior are modeled, and the likelihood model P(x|z) is something that is implicitly defined via Bayes; again, this means that $P(z, z')$ and $Q(z, z' | x, x')$ are **both** optimized. Moreover, the mutual information discussed is over the distribution given by $Q(z, z')$, which is perfectly fine. But what is not fine, is to also optimize $P(z, z')$ to be equal to $Q(z, z')$ and simply remove the KL term -- this effectively removes the prior, and you are left with an autoencoder.
> >
> > Another interpretation to your math is simply that we have an autoencoder with (optimizable) encoder $Q(z, z' | x, x')$ and decoder $P(x, x' | z, z')$ (implicitly defined via Bayes'rule as the posterior from Q and P_data, not $P(z, z')$, see (19)), and we simply define $P(z, z') = \int Q(z, z' | x, x') P_{data}(x, x') d(x, x')$ implicitly as well, such that the KL term becomes zero (according to (30)), and this effectively removes the "prior" from consideration. The moment you try to define / enforce a prior $P(z, z')$ **before optimization of Q**, you have to optimize the MI and KL jointly, and that breaks the statement. I know there are generative models based on variational autoencoders that optimize the $P(z)$ for generative modeling purposes, but that can also be interpreted as training autoencoder first with a flexible model that always perfectly fits the aggregate posterior, so the relationship between x and z are more like autoencoder than variational autoencoder. Again, since P(z) is data-dependent, it is not a prior in the sense that you have already taken data into account, even though the Jensen's inequality still holds.
> >
> > I would highly suggest that the authors revisit the terms that are being optimized, and identify a case where the prior does not depend on the data. Otherwise, this Bayesian interpretation may risk violating Cox's theorem (https://statmodeling.stat.columbia.edu/2016/03/25/28321/).

---

> > > ### Author Response · Authors · 2022-11-18
> > > **In a VAE, we use the data to learn the parameters of the prior + likelihood.  The resulting optimal prior + likelihood depend on all the training data.**
> > >
> > > We updated Sec 3.5 in response to your initial comment to emphasise that we do not set $P(z, z') = Q(z, z')$.  Instead, when we learn the parameters of a sufficiently flexible prior, $P_\theta(z,z')$, we get $P_{\theta^*}(z,z') = Q(z,z')$.
> > >
> > > But the real concern appears to be that using a prior, $p_\theta(z)$ and a likelihood $p_\theta(x|z)$ with parameters $\theta$ and optimising those parameters by maximising either the marginal likelihood or ELBO isn't "truly" Bayesian (apologies in advance if we have misinterpreted).  The reviewer is of course correct.  In the ideal case, we would put priors over all quantities, including over all parameters, $\theta$.  But in many, many cases, Bayesian reasoning over these parameters is intractable, while Bayesian reasoning over the latents is tractable.  In these cases we often choose to integrate over the latent variables, and maximise over the parameters.
> > >
> > > While it isn't truly Bayesian, this is precisely what we do in a VAE.  For instance, see [1], Eqs. 1.13 and 1.14 or Fig. 2.1.  Both the prior, $p_\theta(z)$ and the likelihood, $p_\theta(x|z)$ have parameters $\theta$, where $x$ is the data and $z$ is the latent variable.  To learn the parameters, we combine Eqs. 1.13 and 1.14 from [1] to give the marginal likelihood,
> > > \begin{align}
> > >   P_\theta(x_i) = \int dz_i \; P_\theta(x_i|z_i) P_\theta(z_i).
> > > \end{align}
> > > If we can compute the marginal likelihood analytically, then we find the optimal parameters, $\theta$, by maximising this marginal likelihood.  Of course, in the VAE setting, this marginal likelihood is intractable, so instead we use a lower-bound on the marginal likelihood, the ELBO (see [1] for further details).
> > >
> > > For further context, we do maximize the marginal likelihood in two very common non-VAE settings.  First, expectation maximisation [2] is an algorithm designed to do just this (find the optimal parameters, $\theta$; see e.g. the Wikipedia article en.wikipedia.org/wiki/Expectation–maximization_algorithm).  Second, GP priors are defined by the GP kernel, and the GP kernel has (hyper) parameters such as a length scale.  Strictly speaking, we should do Bayesian inference over these hyperparameters.  However, given the difficulties involved, we usually maximise [3].
> > >
> > > [1] Kingma, D.P. and Welling, M., 2019. An introduction to variational autoencoders. Foundations and Trends® in Machine Learning, 12(4), pp.307-392.
> > >
> > > [2] Dempster, A.P., Laird, N.M. and Rubin, D.B., 1977. Maximum likelihood from incomplete data via the EM algorithm. Journal of the Royal Statistical Society: Series B (Methodological), 39(1), pp.1-22.
> > >
> > > [3] Lalchand, V., Bruinsma, W.P., Burt, D.R. and Rasmussen, C.E., 2022. Sparse Gaussian Process Hyperparameters: Optimize or Integrate?. arXiv preprint arXiv:2211.02476.

---

### Review · Reviewer_cHbB · 2022-11-02

**Summary Of Contributions:**

The paper argues that InfoNCE is a special form of the proposed self-supervised variational autoencoders (SSVAEs)
and uses this relation to explain the success of InfoNCE, which is due to the specific structure prior SSVAEs correspond to.
A synthetic moving ball example is used to show that the prior choice interpretation is useful when the InfoNCE prior fails.

**Audience:**

Yes

**Broader Impact Concerns:**

The work is mostly theoretical, but the author may relate the new connection to some fairness perspectives as the prior interpretation gives the opportunity to improve the fairness of self-supervised learning.

**Claims And Evidence:**

No

**Requested Changes:**


See my points in the weaknesses part in the section above.
Among them, the critical ones are the
- 1.1
- 1.2
- 2.1

and those good to have are
- 1.3
- 2.2


**Strengths And Weaknesses:**

Strengths
- The new connection between InfoNCE and SSVAEs is novel and interesting.
- The newly proposed SSVAEs rely on some novel tricks (implicit definition of likelihood).

Weaknesses
1. The argument of the special choice of prior in SSVAEs contribute to the success of InfoNCE is weak.
    1. More explanation of why the specific prior choice of InfoNCE is useful for representation learning in general is needed.
    2. More explanation of why the corresponding priors of InfoNCE with more powerful mutual information estimators are not encouraged.
    3. More results showing the SSVAE interpretation is needed, e.g. (1) non-deterministic decoder with InfoNCE prior (2)
2. Questions about the moving ball example
    1. It looks to me that (34) is not too different from (36). If I expand (36), I would obtain $\exp(-\frac{1}{2 L^2} (z^\top z + (z')^\top z' - 2z^\top z' ))$, which is similar to (34) with two terms on the norms of $z$ and $z'$ (and a less general "scaling" term shared by all dimensions). I don't understand how these differences would contribute to the success of (36) as the knowledge of "$z'$ is likely to be close to $z$" is represented in a similar manner.
    2. Regarding the point on less general "scaling" term. If we stretch the x-axis by $2$, I would assume the prior with (36) would fail (as we shall not use the same $L$ for both dimensions). Maybe this is another example to demonstrate a correct choice of prior is needed.
3. Typos
    1. SSVAE is not explained in the abstract.

---

> ### Author Response · Authors · 2022-11-03
> **Response**
>
> Thanks for your considered review!
>
> 1.1,1.2) The arguments around why pure MI is undesirable and why the simplified MI estimators corresponding to our priors are more desirable all come from Tschannen et al. (2019).  Their paper is all about this question, and they do a really good job of answering it!  So we are very reluctant repeat too much of that into the Background.  But we have included the key point: that the MI is invariant to arbitrary invertible transformations.  These transformations can arbitrarily entangle the latent variables, so do not encourage good representations.  In contrast, the linear structure in the simplified MI estimator (Eq. 11) might encourage disentangled representations.  We have added more about this to the Background, including a discussion of how the InfoNCE objective only bounds the MI, even in the infinite width limit.
>
> 2.1) $f(z, z') = exp(z^T W z')$ does not encourage $z$ to lie close to $z'$.  It encourages $z = \lambda W z'$, where $\lambda$ is as large as possible.  (Of course, it would encourage proximity if were restricted to lie e.g. on a hypersphere, but we do not have that constraint in this context).
>
> 2.2) Yes, if we expected different behaviour along each axis, we could use a different length scale along each axis.
>
> We have removed references to SSVAE (now CSSVAE) from the abstract.

---

> > ### Comment · Reviewer_cHbB · 2022-12-02
> > **Re-asking question 1.3 after reading the discussion**
> >
> > I read through the thread in https://openreview.net/forum?id=SGNIcTOtvG&noteId=R9ZowkCyr4 and would like to expand question 1.3 in my original reviews a bit here, mostly around the VAE confusion there.
> >
> > 1. If the VAE interpretation is important/necessary, why do we have to strict ourself to a deterministic encoder?
> > - Like you said in https://openreview.net/forum?id=SGNIcTOtvG&noteId=c7cxtvSMNH, as from the original VI motivation, we should obtain better results if we have a better variational approximation, no?
> > - If it has to be a deterministic encoder, I would agree with Reviewer 5HXJ's point that this is more like a AE.
> >   - Re. the point on reconstruction loss: If it's OK to have an implicit decoder for a VAE as you proposed, it's also OK to have a similar implicit decoder for AE. If you plug the deterministic variational posterior into (22), you are still maximizing the log-likelihood defined by the implicit decoder (which would be a reconstruction loss if e.g. a Gaussian decoder is used).
> >
> > 2. How should I understand (27) if I also plug the deterministic variational posterior into $Q_\phi(z)$ and $Q_\phi(z')$?
> > As $Q_\phi(z)$ is defined through $Q_\phi(z \mid x)$ as well, we will have
> > $$
> > Q_\phi(z) = \int Q_\phi(z \mid x) P_\text{true}(x) \mathrm{d}x = P_\text{true}(x)
> > $$
> > and (27) would be come
> > $$
> > \log P_{\theta,\phi}(x,x') = \log \frac{P_{\theta,\phi}(z,z')}{P_\text{true}(x) P_\text{true}(x')} + \text{const}
> > $$
> > Does it simply mean that we are maximizing the joint probability of $z$ and $z'$, for which we have to use a structural prior to encourage them to be similar in some sense?

---

> > > ### Author Response · Authors · 2022-12-02
> > > **Response**
> > >
> > > 1.  We are not restricted to a deterministic encoder.  We have updated the manuscript to emphasise this point even more strongly:
> > >
> > > > We consider fully general encoders encoders, $Q_\phi(x|z)$ and $Q_\phi(x'|z')$ which could be stochastic or deterministic.
> > > All our derivations apply to stochastic or deterministic encoders, except Eq. 27 in Sec. 3.3.
> > >
> > > 2. The first equality you have written is correct, but the second equality isn't.
> > > \begin{align}
> > >   Q_\phi(z) = \int dx Q_\phi(z|x) P_{\rm{true}}(x) \neq P_{\rm{true}}(x)
> > > \end{align}
> > > We can see this by looking at the arguments. $Q_\phi(z|x) P_{\rm{true}}(x)$ has two arguments, $(x,z)$.  When we integrate over $x$, in the middle expression, we are left only with a $z$ argument.

---

> > > > ### Comment · Reviewer_cHbB · 2022-12-02
> > > > **Follow-up**
> > > >
> > > > > We are not restricted to a deterministic encoder. We have updated the manuscript to emphasise this point even more strongly:
> > > >
> > > > I do understand the derivation works for any Q except (27). However, it seems that the non-deterministic encoder was not empirically verified in any experiment, and it's hard to see that why a VAE treatment is necessary here.
> > > >
> > > > > The first equality you have written is correct, but the second equality isn't.
> > > >
> > > > As the decoder is deterministic, you would have $Q_\phi(z \mid x) \neq 0$ only for the $x$ that used to compute $z=g_\phi(x)$ (or multiple $x$ in the case of a many-to-one mapping for $x \to z$. In any case, as the decoder is deterministic, $Q_\phi(z)$ would reduce to $P_\text{true}$ evaluated on one or a few points, without any weighting (due to the fact $Q_\phi(z \mid x)$ is a delta), which is very different from if a non-deterministic is used.

---

> > > > > ### Author Response · Authors · 2022-12-02
> > > > > **Response**
> > > > >
> > > > > The difference between the AE and VAE is not in whether one distribution or the other is deterministic.  Its in what training objective you use.  VAEs use the ELBO.  AEs usually use reconstruction error.  We need the ELBO because we are interested in making a connection to InfoNCE.  To do that we need the ELBO, and once we're using an ELBO, we're doing variational inference / VAEs.  You cannot connect plain AE's to InfoNCE, because the plain AE objective is usually reconstruction error, and InfoNCE doesn't have a reconstruction.  (Also see the comment entitled "VAEs are doing VI, and don't have much to do with AEs (except a loose structural similarity).").
> > > > >
> > > > > We'll take a look at an experiment with a noisy encoder.  However, we expect it to make little if any difference, as a delta function can be obtained by taking a Gaussian distribution with a small variance (formally, taking the limit as the variance goes to zero).  Indeed, if we optimise the noise variance, we expect it to end up being small (at least for an optimised prior) as for an optimised prior, the objective is just the MI, and the MI is never increased by adding noise.
> > > > >
> > > > > In terms of the distributions, we're basically using the deterministic $Q_\phi(z|x)$ to define a change of variables between $x$ under $P_{\rm true}(x)$ and $z$ under $Q_\phi(z)$.  Of course, that change of variables requires a Jacobian term,
> > > > > \begin{align}
> > > > > P_{\rm true}(x) =  |\frac{df}{dx}| Q_\phi(z)
> > > > > \end{align}
> > > > > However, that's only if we have a one-to-one mapping from $x \rightarrow z$.  We don't impose that assumption.  We have a many-to-one mapping which gives something even more complicated, because we need to sum over $x$'s such that $f(x) = z$,
> > > > > \begin{align}
> > > > > Q_\phi(z) = \int dx Q_\phi(z|x) P_{\rm true}(x) = \int dx \delta(z - f(x)) P_{\rm true}(x) = \sum_{x \phantom{a} {\rm st} \phantom{a} z=f(x)} |\frac{df}{dx}|^{-1} P_{\rm true}(x)
> > > > > \end{align}
> > > > > This also means that a deterministic encoder doesn't necessarily imply a deterministic decoder.  In particular, if the encoder is deterministic but many-to-one, then in the decoder, one $z$ could map to different $x$'s.

---

> > > > > > ### Comment · Reviewer_cHbB · 2022-12-02
> > > > > > **Follow-up comment**
> > > > > >
> > > > > > > The difference between the AE and VAE is not in whether one distribution or the other is deterministic. Its in what training objective you use. VAEs use the ELBO. AEs usually use reconstruction error.
> > > > > >
> > > > > > If you set the variational approximation to be a delta function and the decoder is fixed-variance Gaussian, VAE becomes AE and optimizing the *standard* ELBO is equivalent to optimizing the reconstruction error. What would you obtain if you start the `x' <- z' - z -> x` graphical model from this formulation? Please also see more in my comment in the second bullet point of question 1 in https://openreview.net/forum?id=SGNIcTOtvG&noteId=0BuT1OVCEV.
> > > > > >
> > > > > > > We'll take a look at an experiment with a noisy encoder.
> > > > > >
> > > > > > I just want to emphasize that my main point here is to provide some evidence that the VAE perspective is necessary but not optional. As for now the only theory got verified is the one with deterministic decoder, it concerns me that if this VAE perspective is actually necessary or true in general. Experimentation is one way because I'm not convinced by the AE discussion about reconstruction loss above.
> > > > > >
> > > > > > > In terms of the distributions, ...
> > > > > >
> > > > > > Thanks for the explanation here. It makes more sense to me now.

---

> > > > > > > ### Author Response · Authors · 2022-12-03
> > > > > > > **Response**
> > > > > > >
> > > > > > > Fantastic!
> > > > > > >
> > > > > > > One final thought.  There is indeed a close connection between the AE reconstruction loss and the most common VAEs which use a Gaussian likelihood,
> > > > > > > \begin{align}
> > > > > > > \log P(x_i|z) = \mathcal{N}(x_i; \mu_i(z), \sigma^2)
> > > > > > > \end{align}
> > > > > > > where $\mu$ is a neural network.  However, this only happens if we have a Gaussian likelihood.  The VAE framework allows you to use much more general likelihoods [1].  We use this greater flexibility of the VAE framework, by choosing a non-Gaussian likelihood (Eq. 9).  Indeed, we can't even write down an explicit form for our likelihood.  All we have is the implicit form in Eq. (9).
> > > > > > >
> > > > > > > To flip it around, the implicit likelihood in Eq. (9) is not a Gaussian, and therefore cannot be understood as a reconstruction error.  This is fine in the VAE setting, as VAEs can use non-Gaussian likelihoods.  But Eq. (9) is then very difficult to understand in the AE setting, as there is no connection to reconstruction errors.
> > > > > > >
> > > > > > > [1] Kingma, D.P. and Welling, M., 2019. An introduction to variational autoencoders. Foundations and Trends® in Machine Learning, 12(4), pp.307-392.

---

### Author Response · Authors · 2022-11-03
**Overall response to reviewers**

We thank the reviewers for their careful and considered and positive comments.

The reviewers did not identify any substantive issues as regards correctness, though we have made numerous clarifications in response to reviewer comments.

However, the reviewers did highlight the need to clarify the contributions.  In response, we have highlighted two points:
* We provide a connection between perhaps the most important quantity in Bayesian inference (the Bayesian model evidence) and perhaps the most important quantity in information theory (the mutual information).  This establishes a profound connection between Bayesian inference and information theory.  And as highlighted by reviewer 5HXJ, this result is highly unexpected.
* The usual InfoNCE objective with a restrictive choice of $f$ provides only a lower bound on the MI, even in the infinite sample limit (Oord et al. 2018).  In contrast, we show that the infinite-sample InfoNCE objective is actually equal to the log-Bayesian model evidence with any choice of $f$.  Given that in practice $f$ is indeed restricted to a narrow family of functions (Eq. 11), this would suggest that the InfoNCE objective is better understood in terms of Bayesian inference than MI maximisation.

---

### Comment · Reviewer_ZXW3 · 2022-11-24
**Representation learning**

Hi all,

I'm the only one in support of this paper so would like to share my understanding and perhaps clarify some points for the authors. If I made serious mistakes please correct me.

I agree that the derivations in Section 3.2 is strange, but bear in mind that x and x' are not independent draws from the dataset. My understanding confirmed by the authors is that x and x' are dependent because they are patches from the same bigger image. Thus **the graphical model at the beginning of Section 3.2 is reasonable** (though one may draw other versions for such dependence
). This dependence is an important point and I will explain it later.

It is also true that the implicit definition of the decoder (14) is rather unconventional. However, the fact that **the encoder and prior are both optimized or made data-dependent does not mean this approach is not Bayesian**---there should not be any conceptual issue if we just want to capture the data distribution. What is actually strange is that the implicit decoder (14) gives it an arbitrary power to match the data distribution for any distribution over z. The decoder could also ignore z and just be the data distribution. So, let's now assume that the job of density estimation is done: also note that the RHS of (18) and (19) do not involve any x symbols (it is in c); there is really no motivation for the model to learn to generate x from this point onwards. The marginal distribution of the model on x is always the data distribution. **This is impossible for a conventional autoencoder, so any reference made to AE is almost irrelevant.**

Now does this mean this is not a VAE? Perhaps so, and the authors should perhaps correct this interpretation in a major revision. One can say that starting from the ELBO while assuming the powerful decoder (14) is already not a VAE, but this is mainly conceptual: my hot take is that **starting a derivation from the ELBO does not constrain the approach to be a VAE or any generative model, nor does any KL regularizers have to appear anywhere.**

**What's really doing the job of representation learning is Section 3.4**: the authors optimize this objective over dependent pairs x and x'. When P(z, z') = Q(z, z') we get mutual information for posteriors induced by dependent data points (29). Had x and x' been independent, one can even come up with trivial solutions of Q(z|x)=Q(z'|x')=Q(z)=P(z), and P(z, z') = Q(z, z')=Q(z)Q(z'). Nothing needs to be learned based on (25). The objective (25) only makes sense when optimised with dependent pairs, because Q(z, z') != Q(z)Q(z'), and the first KL term of (29) is nonzero and can be maximized to increase mutual information. Again, nothing technical seems wrong to me from this perspective.

The main factors that allowed this derivation from ELBO to MI: optimising the ELBO over dependent pairs of x and x', and the implicit, perfect decoder.

I also want to highlight Section 3.6. If the above content is reasonable, are there serious issues with Section 3.6? If not, **then this is another contribution of this paper that links to InfoNCE**.

Once again, I apologise if I made serious mistakes here.

---

> ### Comment · Reviewer_5HXJ · 2022-11-24
> **Interpretation of the paper**
>
> I actually quite agree with your "hot take", and in fact it is the "VAE" terminology that I had the most problem and confusion with. InfoNCE is a lower bound of mutual information if we assume bounded capacity over $f$, so the additional KL term is reasonable. Perhaps the more precise statement would be "InfoNCE with specific priors is a lower bound of MI, and this gap can be characterized by a KL term. This relationship is conceptually similar to how variational autoencoders (with infinite capacity decoders but finite capacity priors) are lower bounds on likelihood". This view also makes a nice connection (or perhaps even an alternative) to V-information, which uses a variational alternative to mutual information to quantify shared information.
>
> The part with the most confusion is indeed that while VAE is used in most statements, ELBO is the actual quantity being used, and ELBO can mean many things -- for example, we can optimize encoder, decoder and (infinite capacity prior), such that ELBO is simply an autoencoder objective with learnable latent variable distribution.
>
> ---
>
> Another view that does not require two separate set of x's is simply to define $x' := x$ and $Q(z' | x') = $ deterministic encoder to $x'$ (the derivations are only over these symbols, so the math should still hold). Then you end up with
> $$
> E_{Q(z, x)}[\log Q(z, x) - \log Q(z) - \log Q(x)] - KL(Q(x, z) \Vert P(x, z))
> $$
> where the first term is the mutual information between $x, z$ in $Q$ and the second term is supposed to be zero. To make sure second term is zero, we need to 1) make sure that $P(x | z) = Q(x| z)$ (this is by default if no decoder is assumed, and has to optimized if decoder is assumed) and 2) make sure that $P(z) = Q(z)$ (this is via optimizing the so called "prior").
>
> ---
>
> All in all, I do believe the paper has its merits (I don't challenge the math derivations), but I think the interpretation of the math (the writing) can be made clearer. For example, the statement in the abstract that "Under a particular choice of prior we show that the MI is equal to the VAE objective".
> - First of all, this does not type check: MI here is a quantity that depends on data distribution and encoder, but VAE objective is supposedly dependent on also the decoder.
> - Secondly, the decoder in VAE is a key part, but this is implicitly defined during the derivations. At this stage, the reader would be understandably confused about why this is related to VAE if there is no decoder at all.
> - Thirdly, the choice of prior is not trivial, and it is so special that it makes the VAE objective almost like an AE objective.
>
> In fact, I believe that the abstract alone would need a bit of rework. It is not so clear what the main claim is, and there is a tendency of making the important sound less important (the variational interpretation of InfoNCE) and vice versa (representation learning, InfoNCE is a bound on MI).
> An example is on the statement "We offer an alternative motivation for the InfoNCE objective by showing that in the infinite sample limit it is equal to the log-Bayesian model evidence but only bounds the MI.". It makes is sound like the part after "but" is critical, but I would argue that InfoNCE <= MI is quite well known. In fact, there are proofs based purely on the Donsker-Varadhan variational representations of KL divergence (https://arxiv.org/pdf/2007.09852.pdf).

---

> > ### Author Response · Authors · 2022-11-26
> > **VAEs are doing VI, and don't have much to do with AEs (except a loose structural similarity).**
> >
> > Given confusion around basic definitions, and exactly how broad / narrow ELBOs, VI etc. are, I thought I would give some basic background.  Apologies if this is unnecessary.
> >
> > The ELBO, $\mathcal{L}$, is the "Evidence Lower Bound Objective".  Just looking at the name, it needs to be a bound on the model evidence, which is defined as $\log P(x)$.  Of course, the model evidence only makes sense in the context of a probabilistic generative model of the data, $P(x)$, so the ELBO also only makes sense in the context of a probabilistic generative model of the data.
> >
> > So the ELBO implies the existence of generative model (so that it can bound that model's evidence).  The next question is quite how specific the ELBO is: could it be any bound, or is it a specific bound?
> > The ELBO is one and only one bound,
> > \begin{align}
> >   \mathcal{L} = E_{Q(z)}[\log P(x, z) - \log Q(z)].
> > \end{align}
> > which is parameterised in terms of an approximate posterior, $Q(z)$.
> > Of course, we can make different choices about the form for the generative model, $P(x, z)$ and approximate posterior, $Q(z)$.
> > But we cannot get away from the fundamental structure, and the need for a generative model, $P(x, z)$, and an approximate posterior, $Q(z)$.
> >
> > Critically, VI is synonymous with the use of the ELBO.
> > We do VI by optimizing the ELBO, and if we optimize an ELBO, we're doing VI [1].
> >
> > That raises the question of how VAEs relate to VI.
> > VAEs are a special case of VI (so all VAEs are doing VI; see [2]).
> > As VAEs are doing VI, that implies they are using the ELBO in the context of a probabilistic generative model.
> > The key thing that makes general VI into a VAE is that we have a neural network that parameterises our approximate posterior.
> > In particular, in classical VI, the approximate posterior might be e.g. a Gaussian, and we would directly optimize the mean $\mu_i$ and standard deviation, $\sigma_i^2$,
> > \begin{align}
> >   Q(z_i) &= \mathcal{N}(z_i; \mu_i, \sigma_i^2).
> > \end{align}
> > In contrast, in a VAE, we have a neural network that produces the means and variances, and we optimize the weights of the neural network, $w$,
> > \begin{align}
> >   Q(z_i|x_i) &= \mathcal{N}(z_i; \mu_w(x_i), \sigma_w^2(x_i)).
> > \end{align}
> > VAEs therefore should be understood in the context of VI.
> >
> > In contrast, autoencoders are quite different (I would argue the name "VAE" is actually very confusing, and it should really be "Neural Network Amortized VI", but this boat has clearly sailed).
> > VAEs really have very little similarity to autoencoders except for a loose similarity in their architectures.
> > In particular, an autoencoder is simply a model architecture where we take data, $x$, encode it usually deterministically into a low-dimensional latent, $z$, then reconstruct it.
> > We train the encoder and decoder using squared error between the original data and the reconstruction.
> > There isn't any generative model, $P(x, z)$ here, nor is there an approximate posterior, and thus the is no ELBO.
> >
> > Indeed, the Wikipedia page for VAEs (https://en.wikipedia.org/wiki/Variational_autoencoder) emphasises the kinship with variational Bayes, and states that there is only loose structural similarity with AEs:
> > > In machine learning, a variational autoencoder (VAE) ...  belonging to the families of probabilistic graphical models and variational Bayesian methods. Variational autoencoders are often associated with the autoencoder model because of its architectural affinity, but with significant differences in the goal and mathematical formulation.
> >
> > [1] Blei, D.M., Kucukelbir, A. and McAuliffe, J.D., 2017. Variational inference: A review for statisticians. Journal of the American statistical Association, 112(518), pp.859-877.
> >
> > [2] Kingma, D.P. and Welling, M., 2019. An introduction to variational autoencoders. Foundations and Trends® in Machine Learning, 12(4), pp.307-392.

---

> > ### Author Response · Authors · 2022-11-26
> > **Response**
> >
> > > "InfoNCE with specific priors is a lower bound of MI, and this gap can be characterized by a KL term. This relationship is conceptually similar to how variational autoencoders (with infinite capacity decoders but finite capacity priors) are lower bounds on likelihood".
> >
> > We're a bit confused by this as InfoNCE, in the original formulation, doesn't have a prior.  Indeed, Oord et al. 2018 only mentions the term "prior" once in the context of "prior work".  The notion of a prior for InfoNCE only makes sense after accepting our interpretation of the InfoNCE objective as the ELBO for a CSSVAE.
> >
> > > The part with the most confusion is indeed that while VAE is used in most statements, ELBO is the actual quantity being used, and ELBO can mean many things -- for example, we can optimize encoder, decoder and (infinite capacity prior), such that ELBO is simply an autoencoder objective with learnable latent variable distribution.
> >
> > In certain circumstances, the autoencoder objective can be made to resemble the ELBO.  But we're not sure of the relevance of this observation.  We choose families for $P(x, z)$ and $Q(z)$ and optimize the ELBO, so we're definitely doing variational inference.  Any incidental similarity to an AE doesn't mean that we aren't doing VI.  This claim is especially confusing as we really don't see any resemblance to an autoencoder.  In particular, the AE objective typically is something like a squared reconstuction error. We don't do anything like that; indeed, we can't as we don't even have a reconstruction.  Also see our comment entitled "VAEs are doing VI, and don't have much to do with AEs (except a loose structural similarity)."
> >
> > > Another view that does not require two separate set of x's
> >
> > We're not sure of the relevance of this as InfoNCE very definitely requires two sets of x's: e.g. two different augmentations of the same image, or two different timesteps in a time-series.  And we're not sure of the relevance of the later derivations, which involve a mutual information between an observation and a latent variable, as InfoNCE is defined in terms of the MI between two latents.
> >
> > > Under a particular choice of prior we show that the MI is equal to the VAE objective
> >
> > We have replaced this with "We show that when we learn the optimal prior, the VAE objective (the ELBO) becomes equal to the MI."
> >
> > > Secondly, the decoder in VAE is a key part, but this is implicitly defined during the derivations. At this stage, the reader would be understandably confused about why this is related to VAE if there is no decoder at all.
> >
> > This is an interesting point, which we have thought about carefully.  Getting a short, canonical definition of VAEs is difficult.  Perhaps the best we have found is from Wikipedia: "In machine learning, a variational autoencoder (VAE) ...  belonging to the families of probabilistic graphical models and variational Bayesian methods. Variational autoencoders are often associated with the autoencoder model because of its architectural affinity, but with significant differences in the goal and mathematical formulation."  This emphasises the kinship with variational Bayes, and states that the relationship with AEs is just loose structural similarity.  As such, we believe that it is best to define VAEs by starting with VI, and asking what VAEs add.  The answer is that VAE add a neural network approximate posterior.  If we define a VAE as "VI with a neural network approximate posterior", then our CSSVAE is clearly a VAE.
> >
> > > Thirdly, the choice of prior is not trivial, and it is so special that it makes the VAE objective almost like an AE objective.
> >
> > We are very confused as to what looks like an AE objective.  AE objectives are typically e.g. squared reconstruction error, which doesn't make sense in our setting as we don't even have a reconstruction.

---

> > > ### Author Response · Authors · 2022-11-26
> > > **Response**
> > >
> > > > In fact, I believe that the abstract alone would need a bit of rework. It is not so clear what the main claim is, and there is a tendency of making the important sound less important (the variational interpretation of InfoNCE) and vice versa (representation learning, InfoNCE is a bound on MI). An example is on the statement "We offer an alternative motivation for the InfoNCE objective by showing that in the infinite sample limit it is equal to the log-Bayesian model evidence but only bounds the MI.". It makes is sound like the part after "but" is critical, but I would argue that InfoNCE <= MI is quite well known. In fact, there are proofs based purely on the Donsker-Varadhan variational representations of KL divergence (https://arxiv.org/pdf/2007.09852.pdf).
> > >
> > > The main points are:
> > > * We describe an interpretation of InfoNCE in terms of variational inference.
> > > * We show that if we optimise the prior, the log Bayesian model evidence is equal to the MI, establishing a profound connection between Bayesian inference and information theory.
> > > * We argue that the InfoNCE objective is better motivated as arising from our VI approach than as a bound on the MI.  That's because the log Bayesian model evidence is equal to the InfoNCE objective in the infinite sample limit, while the InfoNCE objective forms only a loose lower bound on the MI in the infinte sample limit (as, we agree is well known and was pointed out in Oord et al. 2018).

---

> ### Author Response · Authors · 2022-11-26
> **Response**
>
> > I agree that the derivations in Section 3.2 is strange, but bear in mind that x and x' are not independent draws from the dataset. My understanding confirmed by the authors is that x and x' are dependent because they are patches from the same bigger image. Thus the graphical model at the beginning of Section 3.2 is reasonable (though one may draw other versions for such dependence ). This dependence is an important point and I will explain it later.
>
> Correct.  We think its even clearer if you regard $x$ and $x'$ as part of a timeseries, with $x$ the observation at the previous timestep, and $x'$ as the current observation.  This is in fact the first motivation in the InfoNCE paper.  We have repeated this motivation at relevant places in the manuscript.
>
> > It is also true that the implicit definition of the decoder (14) is rather unconventional. However, the fact that the encoder and prior are both optimized or made data-dependent does not mean this approach is not Bayesian---there should not be any conceptual issue if we just want to capture the data distribution.
>
> Correct.
>
> > What is actually strange is that the implicit decoder (14) gives it an arbitrary power to match the data distribution for any distribution over z. The decoder could also ignore z and just be the data distribution. So, let's now assume that the job of density estimation is done: also note that the RHS of (18) and (19) do not involve any x symbols (it is in c); there is really no motivation for the model to learn to generate x from this point onwards. The marginal distribution of the model on x is always the data distribution.
>
> This not true.  Marginalising $z$ from the model joint (Eq. 12) only gives the true data distribution if the prior over latent is correct, $P(z) = Q(z)$.  Otherwise, mismatch in the prior leads to mismatch in the data distribution.  We have added a note about this under Eq. 12.  Given the rest of the response seems to be based on this, let us know if you want a response to the rest!
>
> > What's really doing the job of representation learning is Section 3.4: the authors optimize this objective over dependent pairs x and x'. When P(z, z') = Q(z, z') we get mutual information for posteriors induced by dependent data points (29). Had x and x' been independent, one can even come up with trivial solutions of Q(z|x)=Q(z'|x')=Q(z)=P(z), and P(z, z') = Q(z, z')=Q(z)Q(z'). Nothing needs to be learned based on (25). The objective (25) only makes sense when optimised with dependent pairs, because Q(z, z') != Q(z)Q(z'), and the first KL term of (29) is nonzero and can be maximized to increase mutual information. Again, nothing technical seems wrong to me from this perspective.
>
> Correct.

---

### Decision · Action_Editors · 2023-01-02

**Recommendation:** Reject

**Comment:**

This is an ambitious and densely written paper that attempts to make progress in an interesting direction. Unfortunately, there was no consensus among the reviewers both on the validity of the claims and the existence of an audience for the paper in its current form. While the authors have improved the manuscript based on some of the reviewer comments, several serious issues still need to be addressed for the paper to be publishable, such as the problems with the claims described above.

The most serious issue is defining the likelihood term in the constructed generative "model" in terms of the true density of the data distribution P_data(x). As this density is unknown, it is hard to call the resulting construct a model. Perhaps P_data(x) can be replaced with a learned component, but then the resulting decoder will only approximate P_data(x) and thus will change the learning signal for the encoder, breaking the equivalence with InfoNCE-based SSL.

The paper is not well written and is unnecessarily hard to follow. For example, is several places, including the abstract, is it not specified that the "VAE" that InfoNCE is claimed to be equivalent to, models the distribution of _pairs_ of (dependent) observations. This is an unfortunate omission because it might lead the readers to think that the "VAE" in question is applied to individual observations, as is typically the case. It is would also be helpful to emphasize the fact that the observation pairs are dependent and that their dependence is the source of the learning signal for representations.

**Audience:**

A clearly established novel connection between contrastive learning using InfoNCE and generative modelling would have been of substantial interest to the community. However, I am not sure there is an audience for the paper in its current form.

**Claims And Evidence:**

The paper makes several problematic claims. The key claim that the paper shows that contrastive SSL methods "implicitly learn a full probabilistic model of the inputs" parameterized as a VAE, does not hold up. The likelihood of this "probabilistic model" is defined as P(x|z) = Q(z|x)P_true(x)/Q(z), which means it depends not only on the intractable aggregate variational posterior Q(z) but also on the unknown true data distribution P_true(x). This means that we can neither evaluate nor sample from the likelihood/model (even using MCMC), so this construction does not define a proper model, not even an implicit one. Calling such a "model" a "VAE" is done in the title of the paper is even less accurate.

The claim that "when we learn the optimal prior, the VAE objective (the ELBO) becomes equal to the MI" is not quite right either because it fails to mention another term in the objective, which does not depend on the encoder parameters.

The claim that "for a deterministic encoder the ELBO is equal to the log Bayesian model evidence" is true, but it is not novel and is not shown rigorously in the paper. The problematic fact that such an encoder results in an infinite KL(q(z|x)||p(z)) term in the ELBO is not acknowledged or addressed. To derive the result rigorously, one can take the limit of the encoder variance going to zero as was done in SurVAE Flows paper [1]. Note that such a model with a deterministic encoder is not a VAE.

The implication in the experimental section that the choice of Gaussian-PDF-like function f which worked well was somehow less natural from the InfoNCE SSL perspective (compared to the advocated generative one) is highly debatable. For example, the difference between using Eq. 39 and Eq. 40 becomes considerably smaller if z and z' in Eq. 39 are restricted to be unit norm, as is often done in self-supervised representation learning. Including this version of f in the very minimal experimental section could provide more evidence for the claim being made.

---

> ### Author Response · Authors · 2023-01-03
> **Response (1/2)**
>
> Reviewer comments
> ========
>
> There was consensus among the reviewers on the validity of the claims.  There was extensive discussion about e.g. connections between VAEs and AEs, and around the "Bayesianness" of fitting parameters of the generative model, but these were resolved.
>
> ZXW3:
> * The main idea of the paper is written clearly, with all derivations and assumptions explicitly stated.
> * The theoretical contribution that links VAE with InfoNCE is correct, novel (to my knowledge), interesting and significant. In particular, the analyses in 3.6 is elegant and give me a good thrill.
> * The experiment clearly makes the point that prior choice is crucial.
>
> cHbB:
> * The new connection between InfoNCE and SSVAEs is novel and interesting.
>
> McsS:
> * Update: the authors have made several changes to the earlier writing which I think clarifies quite a bit, as well as renaming the model they consider. While I still feel this work could benefit from additional rewriting, at this point I would consider it above the bar for publication at TMLR.
>
> 5HXJ:
> * (I don't challenge the math derivations), but I think the interpretation of the math (the writing) can be made clearer
>
> New comments
> ======
>
> Thanks for your extensive further comments on issues that were not originally raised in review.  They all look reasonably straightforward to address in a revision (see below).  Is that option available?
>
> > The paper makes several problematic claims. The key claim that the paper shows that contrastive SSL methods "implicitly learn a full probabilistic model of the inputs" parameterized as a VAE, does not hold up. The likelihood of this "probabilistic model" is defined as P(x|z) = Q(z|x)P_true(x)/Q(z), which means it depends not only on the intractable aggregate variational posterior Q(z) but also on the unknown true data distribution P_true(x). This means that we can neither evaluate nor sample from the likelihood/model (even using MCMC), so this construction does not define a proper model, not even an implicit one. Calling such a "model" a "VAE" is done in the title of the paper is even less accurate.
>
> To begin, we do agree with the reviewers concerns.  However, we chose to run with the "probabilistic model" and "VAE" terminology because of counterarguments that the AC does not appear to have considered.
>
> Critically, if we have toy data, then we might well know $P_{\rm true}(x)$.  And in these toy-data settings where $P_{\rm true}(x)$ is known, we can of course sample and evaluate from the likelihood and model.  Thus, in these toy data settings, everyone should agree that we do indeed implicitly learn a full probabilistic model of the inputs parameterized as a VAE.  Of course, the AC argues that this is not the case in real-data settings where $P_{\rm true}(x)$ is unknown.  Thus, the claim has to be that all our claims are valid if $P_{\rm true}(x)$ is known and invalid if $P_{\rm true}(x)$ is unknown.  But it is not clear that drawing this distinction between known/unknown $P_{\rm true}(x)$ is tenable.  In particular, our fitting procedure does not depend on $P_{\rm true}(x)$, so we always do exactly the same thing, whether $P_{\rm true}(x)$ is known or unknown.  All we rely on is that $P_{\rm true}(x)$ exists, and it does always exist.
>
> This conclusion that all our claims are valid if $P_{\rm true}(x)$ is known and invalid if $P_{\rm true}(x)$ is unknown arises from the definition of a model as something "whose density can be evaluated or sampled".  If we consider a more general and natural definition of "probabilistic model" as "a family of probability distributions that can be fitted to data" then these difficulties vanish. Our approach fits this more general definition.  We define a valid distribution, $P(x)$, that is different from $P_{\rm true}(x)$, and show how to fit $P(x)$ to data.  Sometimes this distribution can be evaluated/sampled.  But not always.  And that's okay (and in fact this is usually the case: we often define distributions for which sampling/ density evaluation is intractable).

---

> ### Author Response · Authors · 2023-01-03
> **Response (2/2)**
>
> > The claim that "when we learn the optimal prior, the VAE objective (the ELBO) becomes equal to the MI" is not quite right either because it fails to mention another term in the objective, which does not depend on the encoder parameters.
>
> The constant is present in all formal derivations, and is mentioned several times in the text.  The only reason that we don't have "(up to a constant)" everywhere is just that its awkward and not very informative.  But this is trivial to fix in a revision.
>
> > The claim that "for a deterministic encoder the ELBO is equal to the log Bayesian model evidence" is true, but it is not novel and is not shown rigorously in the paper. The problematic fact that such an encoder results in an infinite KL(q(z|x)||p(z)) term in the ELBO is not acknowledged or addressed. To derive the result rigorously, one can take the limit of the encoder variance going to zero as was done in SurVAE Flows paper [1]. Note that such a model with a deterministic encoder is not a VAE.
>
> We did not discuss this point in much depth because we thought it was a bit of a side point and that it was obvious in our setting.  In particular, the usual derivation for the ELBO through Jensen gives:
> \begin{align}
>   L = E_Q[\log (P(x, z) / \log Q(z|x))] = E_Q[\log P(x) + \log (P(z|x) / Q(z|x))].
> \end{align}
> In the setting where $Q(z|x)$ is deterministic, we only evaluate the expectation at a point,
> \begin{align}
>    L  = \log P(x) + \log P(z=f(x)|x) / Q(z=f(x)|x)
> \end{align}
> Critically, looking at Eq. 9, $P(z|x)$ is a mixture over particles, with one particle at the deterministic value of $Q(z|x)$.  If you regard P(z|x) and Q(z|x) as discrete distributions over these particles, then you get the (right) answer.  (Though of course we are happy to admit that it is also possible to use a more careful definition involving limits).  It is again trivial to add a discussion of this point including your reference to in a revision.
>
> > The implication in the experimental section that the choice of Gaussian-PDF-like function f which worked well was somehow less natural from the InfoNCE SSL perspective (compared to the advocated generative one) is highly debatable. For example, the difference between using Eq. 39 and Eq. 40 becomes considerably smaller if z and z' in Eq. 39 are restricted to be unit norm, as is often done in self-supervised representation learning. Including this version of f in the very minimal experimental section could provide more evidence for the claim being made.
>
> While restricting to a unit-norm is common, it does not seem particularly well-motivated in a MI setting, as projecting down to a hypersphere will necessarily reduce the MI (as we're pushing z through a many-to-one transformation from a D dimensional Euclidean manifold to a D-1 dimensional manifold).